# Signatures of magnetism control by flow of angular momentum

L. Chen[1✉], Y. Sun[1], S. Mankovsky[2], T. N. G. Meier[1], M. Kronseder[3], C. Sun[4,5], A. Orekhov[4], H. Ebert[2], D. Weiss[3] & C. H. Back[1,6,7]

Exploring new strategies to manipulate the order parameter of magnetic materials by electrical means is of great importance not only for advancing our understanding of fundamental magnetism but also for unlocking potential applications. A well-established concept uses gate voltages to control magnetic properties by modulating the carrier population in a capacitor structure[1–5]. Here we show that, in Pt/Al/Fe/GaAs(001) multilayers, the application of an in-plane charge current in Pt leads to a shift in the ferromagnetic resonance field depending on the microwave frequency when the Fe film is sufficiently thin. The experimental observation is interpreted as a current-induced modification of the magnetocrystalline anisotropy $\Delta H_A$ of Fe. We show that (1) $\Delta H_A$ decreases with increasing Fe film thickness and is connected to the damping-like torque; and (2) $\Delta H_A$ depends not only on the polarity of charge current but also on the magnetization direction, that is, $\Delta H_A$ has an opposite sign when the magnetization direction is reversed. The symmetry of the modification is consistent with a current-induced spin[6–8] and/or orbit[9–13] accumulation, which, respectively, act on the spin and/or orbit component of the magnetization. In this study, as Pt is regarded as a typical spin current source[6,14], the spin current can play a dominant part. The control of magnetism by a spin current results from the modified exchange splitting of the majority and minority spin bands, providing functionality that was previously unknown and could be useful in advanced spintronic devices.

Spin torque (spin-transfer torque and spin–orbit torque), which involves the use of angular momentum generated by partially or purely spin-polarized currents, is a well-known means for manipulating the dynamic properties of magnetic materials. In structures such as giant magnetoresistance or tunnel magnetoresistance junctions, the flow of a spin-polarized electric current through the junction imparts spin-transfer torques on the magnetization in the free ferromagnetic layer[15–17]. In heavy metal (HM)/ferromagnet (FM) bilayers, a charge current flowing in HM induces a spin accumulation at the HM/FM interface and generates spin–orbit torques (SOTs) acting on FM[18]. These torques serve as versatile control mechanisms for magnetization dynamics, such as magnetization switching[19,20], domain wall motion[21–23], magnetization relaxation[24] and auto-oscillations of the magnetization[25,26]. These innovative approaches and their combinations open up a spectrum of possibilities for tailoring magnetic properties with potential implications for technologies such as magnetic random access memories[18,27].

## General considerations

Although the impact of spin currents on the orientation of the magnetization **M** is widely recognized, there have been only a few explicit observations of successful spin-current-driven manipulation of the magnitude of **M**. Previous work[28] has shown that, in a magnetic Ni/Ru/Fe tri-layer in which the two magnetization layers are coupled by an exchange coupling, ultrafast laser-generated super-diffusive spin currents in Ni transiently enhance the magnetization of Fe when the two ferromagnetic layers are aligned parallel and decrease when the two ferromagnetic layers are aligned antiparallel, respectively. This transient effect is limited to low optical excitations because super-diffusive spin currents saturate at high power. To explore the modulation of magnetism by spin current, Fig. 1 shows the process of spin current transfer[15–17,29–31]. Spin accumulation, generated by a charge current **I**, contains both transverse and longitudinal spin components with respect to **M**. It can be generated by the strong spin splitting of the energy band of ferromagnetic metals[17], by spin Hall effect[6], by orbital Hall effect and by subsequent conversion of the orbital current into a spin current by the spin–orbit interaction in the bulk[7] as well as by spin Rashba–Edelstein effect (alternatively named inverse spin galvanic effect)[8] at the interfaces. The incident transverse spin current dephases and is absorbed by **M**, which gives rise to damping-like spin torques and is responsible for the change in **M** direction[29,30]. After spin transfer and in the spin diffusion length of FM, the exiting spin current is on average aligned with **M**, and the spin-up electron can fill the majority band when **M** is along the +**z** direction (Fig. 1a). Owing to the enhanced exchange splitting of the majority and minority spin bands, this leads to an enhancement of $M$ as well as an increase in the magnetic

[1]Department of Physics, Technical University of Munich, Munich, Germany. [2]Department of Chemistry, Ludwig Maximilian University, Munich, Germany. [3]Institute of Experimental and Applied Physics, University of Regensburg, Regensburg, Germany. [4]Department of Chemistry, Technical University of Munich, Munich, Germany. [5]TUMint.Energy Research, Department of Chemistry, Technical University of Munich, Munich, Germany. [6]Munich Center for Quantum Science and Technology, Munich, Germany. [7]Center for Quantum Engineering, Technical University of Munich, Munich, Germany. ✉e-mail: lin0.chen@tum.de

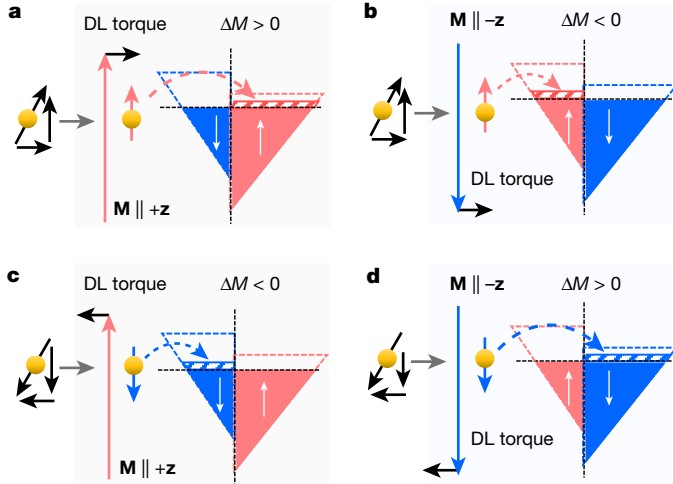

**Fig. 1 | Schematic of the microscopic mechanism of manipulation of magnetism by a spin current. a**, The electron spins transmitted into the FM contain both transverse and longitudinal components with respect to **M**. Owing to exchange coupling, the transverse component dephases and is absorbed by **M**, which gives rise to the damping-like (DL) SOT and is responsible for changing the direction of **M**. The longitudinal component of the spin current is on average aligned with **M**, leading to additional filling of the majority band when **M** is oriented along the +**z** direction, and an enhancement of the magnitude $M$ as well as an increase in magnetic anisotropies are expected because of the enhanced exchange splitting of the majority and minority spin energy bands. **b**, When **M** is aligned along the −**z** direction, the spin-polarized electron enters the minority band, which can lead to a decrease in $M$ as well as a decrease in the magnetic anisotropies because of the reduction in the exchange splitting of the majority and minority spin energy bands. **c,d**, The same as **a** and **b** but the polarization of the spin current is reversed, which is expected to reduce $M$ for **M** ∥ +**z** (**c**) and enhance $M$ for **M** ∥ −**z** (**d**).

anisotropies. When **M** is along the −**z** direction as shown in Fig. 1b, a decrease in $M$ is expected because of the filling of the minority band and the reduction of the exchange splitting. Similarly, once the polarity of the spin current is reversed by reversing the polarity of **I**, a decrease or an increase in $M$ is expected if **M** ∥ +**z** or **M** ∥ −**z**, respectively, as shown in Fig. 1c,d. Therefore, the change in magnetization $\Delta M$ by a spin current is expected to be odd with respect to the inversion of either **I** or **M**, that is, $\Delta M(\mathbf{I}, \mathbf{M}) = -\Delta M(-\mathbf{I}, \mathbf{M}) = -\Delta M(\mathbf{I}, -\mathbf{M})$.

## Ferromagnetic resonance measurements

To prove the above scenario, Pt (6 nm)/Al (1.5 nm)/Fe ($t_{Fe}$ = 4.5, 2.8, 2.2 and 1.2 nm) multilayers with different Fe thicknesses $t_{Fe}$ are grown on a single 2-inch semi-insulating GaAs(001) wafer by molecular-beam epitaxy (Fig. 2b, Methods and Supplementary Note 1). The ultrathin Fe films on GaAs(001) allow us to investigate the expected modification of the magnetic properties for two reasons: Fe/GaAs(001) shows (1) very low Gilbert damping $\alpha$ values in the sub-nanometre thickness regime ($\alpha$ = 0.0076 for $t_{Fe}$ = 0.91 nm) (ref. 32), and thus it is possible to detect the magnetization dynamics for ultrathin samples and (2) strong interfacial in-plane uniaxial magnetic anisotropy (UMA), which is advantageous for the detection of the spin-current-induced modification of magnetic anisotropies. The UMA originates from the anisotropic bonding between Fe and As atoms at the GaAs(001) surface[33], in which the ⟨110⟩ orientations are the magnetic easy axis (EA) and the ⟨Ī10⟩ orientations are the magnetic hard axis (HA) (Fig. 2c). We perform time-resolved magneto-optical Kerr microscopy measurements with out-of-plane driving field to characterize both the static and dynamic magnetic properties of Fe under the influence of spin currents generated by applying a charge current in Pt (Fig. 2a and Methods).

Typical ferromagnetic resonance (FMR) spectra for $t_{Fe}$ = 2.2 nm and for **I** ∥ [110] are shown in Fig. 2d. A clear modification of the FMR spectrum is observed. By fitting the curves with the combination of symmetric and an anti-symmetric Lorentzian (Methods), the resonance field $H_R$ and the full width at half maximum $\Delta H$ are obtained.

## Modification of the linewidth

The dependence of $\Delta H$ on $I$ for $\varphi_{I-H}$ = ±90° is shown in Fig. 2e, where $\varphi_{I-H}$ is the angle between **I** and the magnetic field **H** (Fig. 2d, inset). A linear behaviour with opposite slopes for $\varphi_{I-H}$ = ±90° shows the presence of the damping-like SOT, confirming previous reports[24]. To extract the modification of the linewidth, the $I$ dependence of $\Delta H$ is fitted by

$$\Delta H = \Delta H_0 + [d(\Delta H)/dI]I + c_1 I^2. \tag{1}$$

Here $\Delta H_0$ is $\Delta H$ for $I$ = 0, $d(\Delta H)/dI$ quantifies the modification of linewidth by the spin current and $c_1$ accounts for possible Joule heating effects on $\Delta H$. A detailed measurement of $d(\Delta H)/dI$ as a function of $\varphi_{I-H}$ shows that $d(\Delta H)/dI$ varies strongly around HA. The angular dependence can be well fitted by considering an effective damping-like SOT efficiency $\xi$ of 0.06 (Methods). The weaker damping-like torque, generated by the Bychkov–Rashba-like and Dresselhaus-like spin–orbit interactions at the Fe/GaAs interface, plays a negligible part in the linewidth modulation[34]. As the angular dependence of $d(\Delta H)/dI$ can be well fitted by conventional SOTs[18,35], that is, equation (10) in Methods, there is no need to consider other higher order SOTs[36].

## Modification of the ferromagnetic resonance field

Having identified the modification of $\Delta H$, we now focus on the modification of $H_R$, which is related to the magnetization and magnetic anisotropies. Figure 3a,b shows the $I$ dependence of $H_R$ for $t_{Fe}$ = 2.8 nm measured at selected frequencies $f$ for $H$ applied along EA and HA to avoid magnetic dragging effects[32,34]. As shown at the top of each panel, the current is applied along the [100] orientation, and the direction of spin **σ** is along the [010] orientation with equal projections onto the [110] and [Ī10] orientations. Therefore, this geometry allows a precise comparison of the current-induced modification of $H_R$ between the [110] and [Ī10] orientations in the same device. For **M** ∥ [110] (Fig. 3a), all the $H_R$-$I$ traces show a positive curvature, whereas for **M** ∥ [Ī10] (Fig. 3b), traces with a negative curvature are observed. The positive and negative curvatures along [110] and [Ī10] orientations are because Joule heating reduces the magnetization and thus the UMA, resulting in an increase in $H_R$ along [110] but a decrease in $H_R$ along [Ī10]. Apart from the symmetric parabolic dependence induced by Joule heating, a linear component in the $I$ dependence of $H_R$ is also observed because $H_R(-I) \neq H_R(+I)$ holds. Note that for **M** along both EA and HA, $H_R(-I) > H_R(+I)$ holds for all frequencies. As $t_{Fe}$ is reduced to 1.2 nm, the $I$ dependence of $H_R$ along the EA is similar to the one with $t_{Fe}$ = 2.8 nm and $H_R(-I) > H_R(+I)$ still holds (Fig. 3c). However, for **M** ∥ [Ī10] (Fig. 3d), the relative magnitude of $H_R(-I)$ and $H_R(+I)$ strongly depends on $f$, that is, $H_R(-I) < H_R(+I)$ holds for $f$ = 12.0 GHz; $H_R(-I) \approx H_R(+I)$ holds for $f$ = 14.0 GHz but $H_R(-I) > H_R(+I)$ holds for $f$ = 16.0 GHz. The frequency-dependent shift of $H_R$ indicates that the magnetic properties of Fe are modified by the spin current for thinner samples, an observation that has not been reported before, to our knowledge.

## Modification of the magnetic anisotropies

To quantify the modification of the magnetic anisotropies, the $I$ dependence of the $H_R$ trace is fitted by

$$H_R = H_{R0} + (dH_R/dI)I + c_2 I^2. \tag{2}$$

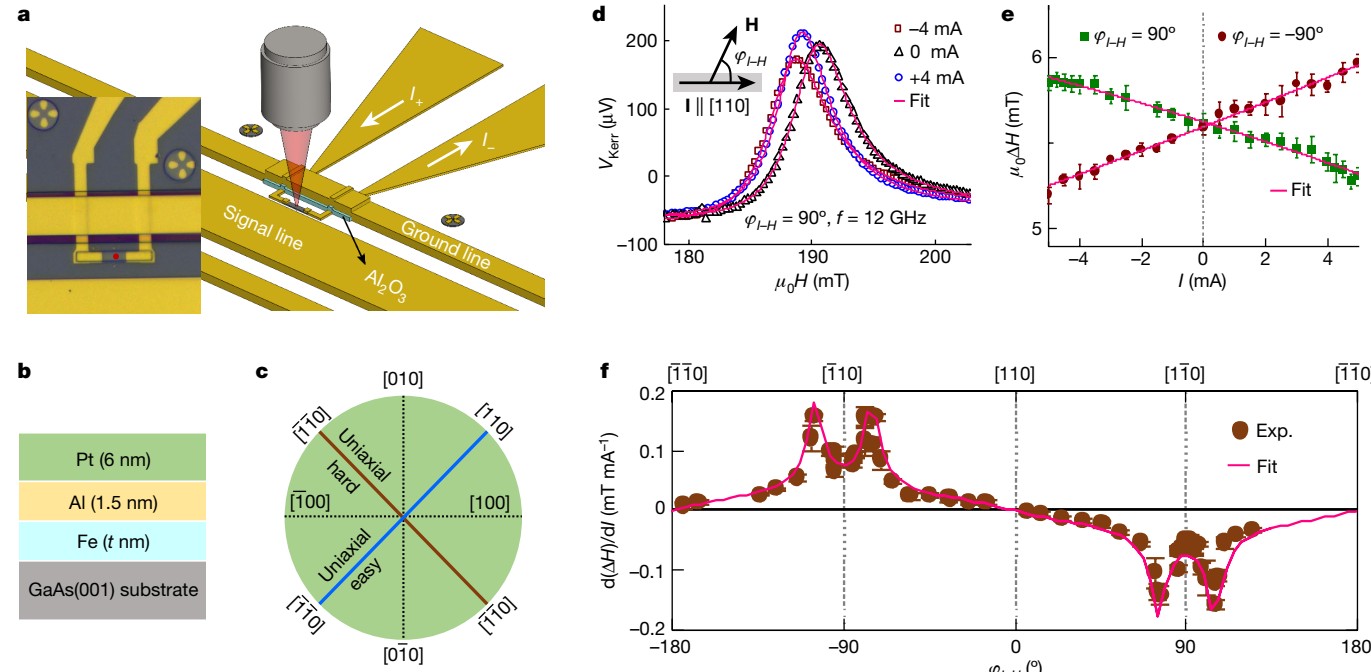

**Fig. 2 | Measurement set-up, device and modification of linewidth by charge current. a**, Schematic of the device for the detection of ferromagnetic resonance by time-resolved magneto-optical Kerr microscopy. **b**, Schematic of the Pt/Al/Fe/GaAs(001) structure. **c**, Diagram of crystallographic axes with EA and HA along the ⟨110⟩ and ⟨Ī10⟩ orientations. **d**, FMR spectra for different d.c. currents $I$ measured at $f = 12$ GHz and $\varphi_{I-H} = 90°$, where $\varphi_{I-H}$ is the angle between the magnetic field and the current direction as shown in the inset. The solid lines are the fits. **e**, FMR linewidth as a function of d.c. current for $\varphi_{I-H} = \pm90°$; solid lines are the linear fits from which the modulation amplitude d($\Delta H$)/d$I$ is obtained. Error bars represent the standard error of the least squares fit of the $V_{Kerr}(H)$ traces in **d**. **f**, $\varphi_{I-H}$ dependence of d($\Delta H$)/d$I$. Error bars represent the standard error of the least squares fit of the $I$–$\Delta H$ traces in **e**. The solid line is the calculated result when taking into account the in-plane magnetic anisotropies of Fe (see Methods).

Here $H_{R0}$ is $H_R$ at $I = 0$, d$H_R$/d$I$ quantifies the modification of $H_R$, and $c_2$ accounts for Joule heating effects on $H_R$. The $f$ dependences of d$H_R$/d$I$ for different orientations of **M** and different $t_{Fe}$ are summarized in Fig. 4. For $t_{Fe} = 2.8$ nm and **M** ∥ ⟨110⟩ orientations (Fig. 4a), d$H_R$/d$I$ is independent of frequency with a positive zero-frequency intercept (about 0.08 mT mA⁻¹) for **M** ∥ [Ī10]. As **M** is rotated by 180° to the [110] orientation, the sign of the intercept changes to negative with the same amplitude as the [Ī10] orientation (around −0.08 mT mA⁻¹). This can be understood in terms of the current-induced Oersted field and/or field-like torque $h_{Oe/FL}$, arising from the current flowing in Pt and Al, which shifts $H_R$. The field-like torque originates from the incomplete dephasing (non-transmitted and/or non-dephased) component of the incoming spin[29,30,37]. For **M** along HA (Fig. 4b), the $f$-independent d$H_R$/d$I$ has also opposite zero-frequency intercepts for **M** ∥ [Ī10] and **M** ∥ [1Ī0] with virtually identical $h_{Oe/FL}$ value as EA. This confirms that the spin accumulation **σ** has equal projection onto the ⟨110⟩ and ⟨Ī10⟩ orientations. As $t_{Fe}$ is reduced to 1.2 nm (Fig. 4c), the intercept of the $f$-independent d$H_R$/d$I$ traces along [110] and [Ī10] orientations, respectively, increases to about −0.20 mT mA⁻¹ and about 0.20 mT mA⁻¹, respectively. However, as **M** is aligned along HA (Fig. 4d), the d$H_R$/d$I$ trace differs significantly from other traces: (1) it is no longer $f$-independent but shows a linear dependence on $f$ with opposite slopes for **M** along the [Ī10] and [1Ī0] orientations, (2) the absolute value of the zero-frequency intercept along HA (about 0.32 mT mA⁻¹) is no longer equal to that along EA (about 0.2 mT mA⁻¹). The $f$ dependence of the d$H_R$/d$I$ traces cannot be interpreted to arise from the frequency-independent $h_{Oe/FL}$ and can be explained only by a change in the magnetic anisotropies induced by the spin current.

In the presence of the in-plane magnetocrystalline anisotropies, the dependencies of $H_R$ on $f$ along EA $H_R^{EA}$ and HA $H_R^{HA}$ are given by the modified Kittel formula[34]

$$\begin{cases} \left(\dfrac{2\pi f}{\gamma}\right)^2 = \mu_0^2\left(H_R^{EA} + H_K + \dfrac{H_B}{2}\right)(H_R^{EA} - H_B - H_U) \\ \left(\dfrac{2\pi f}{\gamma}\right)^2 = \mu_0^2\left(H_R^{HA} + H_K + \dfrac{H_B}{2} - H_U\right)(H_R^{HA} - H_B + H_U), \end{cases} \quad (3)$$

where $\gamma$ is the gyromagnetic ratio, $H_K$ is the effective magnetic anisotropy field due to the demagnetization field along ⟨001⟩, $H_B$ is the biaxial magnetic anisotropy field along ⟨100⟩ and $H_U$ is the in-plane UMA field along ⟨110⟩. The magnitude of $H_K$, $H_U$ and $H_B$ at $I = 0$ for each $t_{Fe}$ is quantified by the angle and frequency dependencies of $H_R$ (Methods). Obviously, a change in the magnetic anisotropy fields $H_A$ ($H_A = H_K$, $H_U$, $H_B$) by $\Delta H_A$ ($\Delta H_A = \Delta H_K$, $\Delta H_U$, $\Delta H_B$) leads to a shift of $H_R$ and the magnitude of the shift $\Delta H_R$, defined as $\Delta H_R = H_R(H_A) - H_R(H_A + \Delta H_A)$, depends on $f$. In the measured frequency range (10 GHz < $f$ < 20 GHz), the $\Delta H_R$–$f$ relations induced by $\Delta H_A$ can be calculated by equation (3), and their dependencies on $f$ are summarized in Extended Data Table 1.

As $h_{Oe/FL}$ also shifts $H_R$ along EA and HA by $\pm\frac{\sqrt{2}}{2}h_{Oe/FL}$, where '+' corresponds to the [110] and [1Ī0] directions, and '−' corresponds to the [Ī10] and [1Ī0] directions, the total $\Delta H_R$ along EA and HA is given by

$$\begin{cases} \Delta H_R^{EA}(f) = \Delta H_U - \Delta H_B \pm \dfrac{\sqrt{2}}{2}h_{Oe/FL} + (k_K + k_B - k_U)f \\ \Delta H_R^{HA}(f) = -(\Delta H_U + \Delta H_B) \pm \dfrac{\sqrt{2}}{2}h_{Oe/FL} + (k_K + k_B)f. \end{cases} \quad (4)$$

Here the slope $k$ [$k = k_K$, $k_U$, $k_B$ and $k = \frac{d(\Delta H_R)}{df}$] quantifies the modulation of $H_R$ induced by $\Delta H_A$. As the $f$ dependence of $\Delta H_R^{EA}$ induced by $\Delta H_U$ has an opposite slope as those induced by $\Delta H_K$ and $\Delta H_B$ (Methods), it is

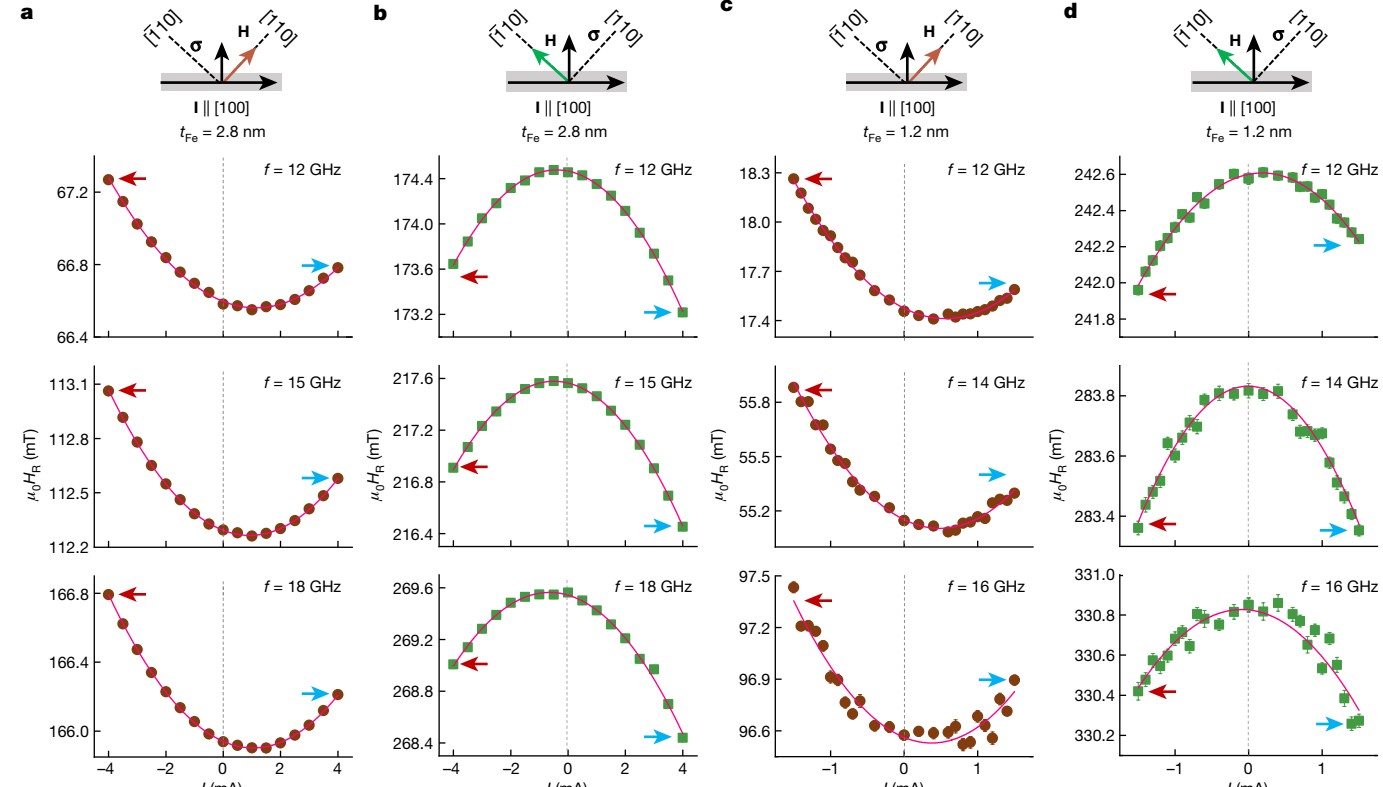

**Fig. 3 | Modification of resonance field. a**, $I$ dependence of $H_R$ measured at selected frequencies for $H$ along [110] for $t_{Fe}$ = 2.8 nm. **b**, The same as **a** but for $H$ along [$\bar{1}$10]. **c**,**d**, The same plots as **a** and **b** but for $t_{Fe}$ = 1.2 nm. Error bars represent the standard error of the least squares fit of the $V_{Kerr}(H)$ traces. The red and blue arrows in each panel are marked to show the relative amplitude of $H_R(-I)$ and $H_R(+I)$. As shown in the top panels, for all the devices, the charge

currents are applied along the [100] orientation, and the direction of the spin accumulation $\sigma$ is along the [010] direction with equal projections onto the [110] and [$\bar{1}$10] orientations. This experimental trick allows an accurate comparison of the current-induced modification for the [110] and [$\bar{1}$10] orientations in the same device.

possible to obtain an $f$-independent $\Delta H_R^{EA}$ along EA by tuning the corresponding parameters and to obtain an $f$-linear $\Delta H_R^{HA}$ along HA. To reproduce the results along the [110] and [1$\bar{1}$0] orientations (that is, the net magnetization of these two orientations is parallel to **I**), we obtain $\Delta H_B$ = 0.26 mT mA$^{-1}$, $\Delta H_K$ = 2.0 mT mA$^{-1}$ and $\Delta H_U$ = 2.5 mT mA$^{-1}$ through equations (3) and (4) (Methods). By contrast, for the datasets for **M** along the [$\bar{1}\bar{1}$0] and [$\bar{1}$10] orientations (that is, the magnetization is rotated by 180° and the net magnetization is antiparallel to **I**), $\Delta H_B$ = −0.26 mT mA$^{-1}$, $\Delta H_K$ = −2.0 mT mA$^{-1}$ and $\Delta H_U$ = −2.5 mT mA$^{-1}$ are obtained, which have the opposite polarity compared with that of **M** along the [110] and [1$\bar{1}$0] orientations.

Figure 4e shows the obtained $\Delta H_A$ as a function of $t_{Fe}$. For $t_{Fe}$ above 2.8 nm, the modification of the magnetic anisotropy is too small to be observed. For $t_{Fe}$ below 2.2 nm, $\Delta H_A$ increases as $t_{Fe}$ decreases. This indicates that the spin-current-induced modification of the magnetic energy landscape is of interfacial origin, similar to the damping-like spin torque determined by the $f$ dependence of d($\Delta H$)/d$I$ (Methods), and a possible magnetic proximity effect has no role in the modification (Supplementary Note 3). The modification changes sign when **M** is rotated by 180°, which fully validates the scenario of $\Delta H_A(\mathbf{I}, \mathbf{M})$ = $-\Delta H_A(-\mathbf{I}, \mathbf{M}) = -\Delta H_A(\mathbf{I}, -\mathbf{M})$ as suggested in Fig. 1. For a given **M** direction, the obtained $\Delta H_B$, $\Delta H_K$ and $\Delta H_U$ have the same sign, which is also consistent with a monotonic increase or decrease in $H_B$, $H_K$ and $H_U$ as temperature decreases or increases, respectively (Supplementary Fig. 7). Moreover, these results also show that $H_U$ is more sensitive to spin current than $H_K$ and $H_B$, highlighting the importance of UMA to enable the observation. The much smaller $\Delta H_B$ value is because $H_B$ is one to two orders of magnitude smaller than $H_U$ and $H_K$ in the ultrathin

regime (Methods). It should be noted that, besides the modification of anisotropy, an anisotropic modification of $\gamma$ could, in principle, explain the experimental results according to equation (3). However, as it is not clear why a modification of $g$ could be anisotropic, we ignore this effect here (Methods).

## Discussions of possible mechanisms

As $H_K \approx M$ holds in the ultrathin regime (Methods), $\Delta H_K$ is directly related to $\Delta M$. The change in magnetization can be attributed to the additional filling of the electronic $d$-band. To a first-order approximation, the filling of the $d$-band by spin current leads to a change in the magnetic moment of the order of $n_s/n_{Fe} \approx 0.16\%$, where $n_s$ is the transferred areal spin density, and $n_{Fe}$ is the areal density of the magnetic moment of Fe. This estimation agrees with the ratio between $\Delta H_K$ and $H_K$, that is, $\Delta H_K/H_K \approx 0.2\%$ (Methods).

By contrast, to mimic the effect of spin current on the UMA and magnetic moment, we have investigated the dependence of the UMA on the external magnetic field by first-principles electronic band structure calculations. The resulting modification of UMA has been determined using magnetic torque calculations[38] (Supplementary Note 4). The applied **H** results in an increase in the magnetic anisotropy energy, if **H** is parallel to **M** and to a decrease in anisotropy in the case of antiparallel orientation. These changes are accompanied by an increase (for $H > 0$) or decrease (for $H < 0$) of magnetic moment, consistent with experimental observations. Moreover, to model a change in $\Delta H_U$ of 2.5 mT for a d.c. current of 1 mA as observed in the experiment, an equivalent magnetic field of about 1.5 T is needed

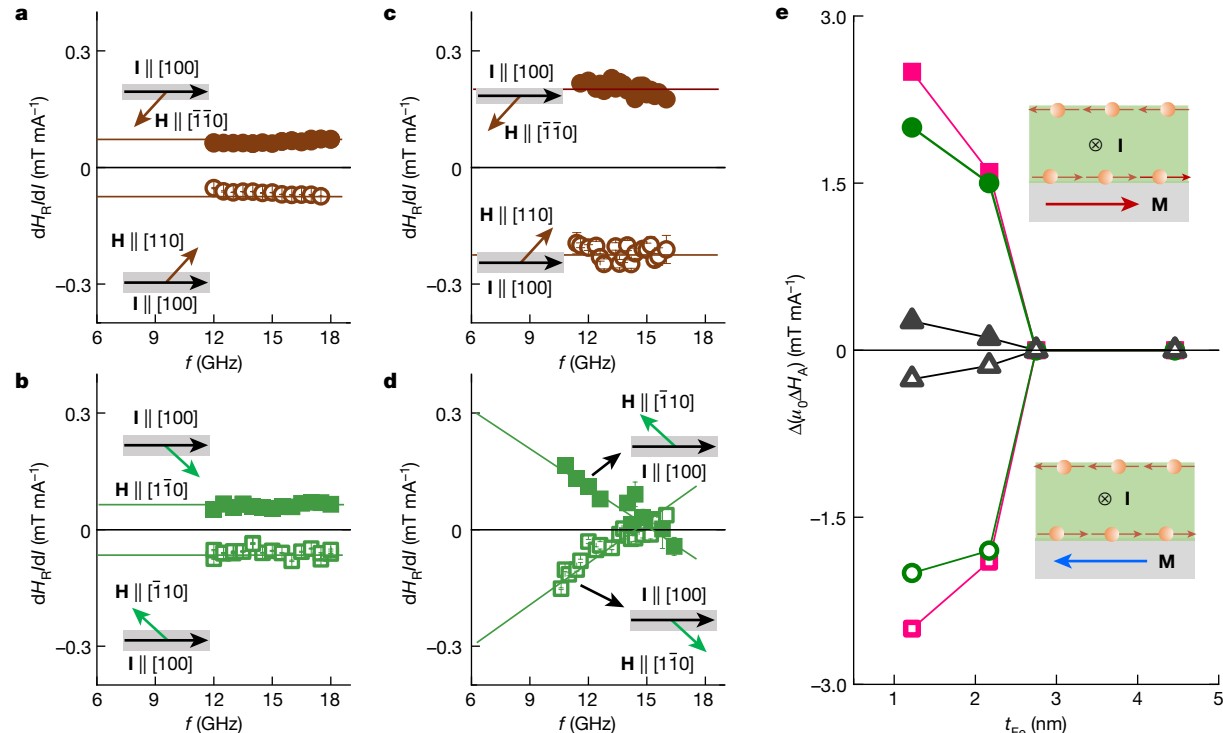

**Fig. 4 | Modification of magnetic anisotropies. a**, The $f$ dependence of $dH_R/dI$ for $H$ along the EA ([110] and [$\bar{1}\bar{1}0$] orientations). **b**, The $f$ dependence of $dH_R/dI$ for $H$ along the HA ([$\bar{1}10$] and [$1\bar{1}0$] orientations). The results in **a** and **b** are obtained for $t_{Fe}$ = 2.8 nm. **c,d**, Same plots as in **a** and **b** but for $t_{Fe}$ = 1.2 nm. Error bars in each figure (most of them are smaller than the symbol size) represent the standard error of the least squares fit of the $I$–$H_R$ traces in Fig. 3. The insets show the relative orientations between the current (**I** ∥ [100], black arrows) and the magnetic field (or magnetization), in which the EA are represented by brown arrows and the HA are represented by green arrows. **e**, Summary of $t_{Fe}$ dependence of $\Delta H_A$ ($\Delta H_A = \Delta H_K$, $\Delta H_U$, $\Delta H_B$) for opposite magnetization **M** directions, in which the solid symbols represent the **M** direction and the open symbols represent the −**M** direction. The inset shows the relative orientations between the charge current **I** and **M**.

(Supplementary Note 4). More sophisticated models might be needed to extend the existing model and to explain the experimental results quantitatively.

## Perspective on spintronics and orbitronics

Our results have shown that the intrinsic properties of ultrathin ferromagnetic materials, that is, the magnitude of $M$ and $H_A$, can be varied in a controlled way by spin currents, which has been ignored in the spin-transfer physics. This unique route of controlling magnetic anisotropies is not accessible by other existing ways using electric field[1–5] and mechanical stress[39,40] in which the control of magnetism is independent of the magnetization direction. Besides the magnitude of the magnetization, other material parameters, such as the Curie temperature and coercive, are also expected to be controllable by spin current. Spin torque plays an essential part in modern spintronic devices; thus, beyond this proof of principle, the so far unnoticed modification of the length of **M** by spin currents could offer an alternative and attractive generic actuation mechanism for the spin-torque phenomena. We expect such a modification of the magnetic energy landscape to be a general feature, not limited to ferromagnetic metal/heavy metal systems with strong spin–orbit interaction but also to be present in the case of conventional spin-transfer torques, in which it is generally believed that the magnitude of **M** is fixed during the spin-transfer process[15–17]. Although the modulation of magnetism is demonstrated by using a single-crystalline ferromagnet, this concept also applies to polycrystalline ferromagnets, for example, Py. Moreover, the modification is not limited to in-plane ferromagnets, and we could manipulate ferromagnets with perpendicular anisotropy by using out-of-plane polarized spin current sources, for example,

WTe$_2$ (ref. 41), RuO$_2$ (refs. 42–45), Mn$_3$Sn (ref. 46) and Mn$_3$Ga (ref. 47). We believe that much larger modification amplitudes can be realized in other more effective spin current sources based on the wide range of spin-torque material choices[18].

Apart from the spin effect mentioned above, recent experimental and theoretical studies have shown that the orbital Hall effect[9] and the orbital Rashba–Edelstein effect[10–12] can generate orbital angular momenta in the bulk of nonmagnetic layers and at interfaces with broken inversion symmetry. The generated orbital momenta can exert a torque on **M** and could also cause a modification of **M** in two ways: (1) the orbital current diffuses into an adjacent magnetic layer and is converted into a spin current by spin–orbit interaction[13,14]. In this case, the modification of **M** is analogous to the scenario discussed for a spin current. (2) The orbital current could, in principle, act directly on the orbital part of **M**, generating orbital torques as well as leading to a modification of the orbital magnetization. The change in **M** by an orbital current is expected to have the same odd symmetry as that induced by a spin current. Importantly, orbital effects could induce an even larger modification than spin effects because of the giant orbital Hall conductivity[9] observed in some materials and could affect thicker ferromagnets as it has been predicted that the orbital current dephasing length is longer than the spin dephasing length[48].

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

## Methods

### Sample preparation

Samples with various Fe thicknesses $t_{Fe}$ are grown by molecular-beam epitaxy (MBE). First, a GaAs buffer layer of 100 nm is grown in a III–V MBE. After that the substrate (semi-insulating wafer, which has a resistivity $\rho$ between $1.72 \times 10^8\,\Omega$ cm and $2.16 \times 10^8\,\Omega$ cm) is transferred to a metal MBE without breaking the vacuum for the growth of the metal layers. For a better comparison of the physical properties of different samples, various Fe thicknesses are grown on a single two-inch wafer by stepping the main shadow shutter of the metal MBE. After the growth of the step-wedged Fe film, 1.5-nm Al/6-nm Pt layers are deposited on the whole wafer. Sharp reflection high-energy electron diffraction patterns have been observed after the growth of each layer (Supplementary Note 1), which indicate the epitaxial growth mode as well as good surface (interface) flatness. High-resolution transmission electron microscopy measurements (Supplementary Note 1) show that (1) all the layers are crystalline and (2) there is diffusion of Al into Pt but no significant Al–Fe and Pt–Fe interdiffusion. Therefore, the magnetic proximity effect between Fe and Pt is reduced. The intermixed Pt–Al alloy can be a good spin current generator. Previous work[49] has shown that alloying Pt with Al enhances the spin-torque efficiency.

### Device fabrication

First, Pt/Al/Fe stripes with a dimension of 4 µm × 20 µm and with the long side along the [110] and [100] orientations are defined by a mask-free writer and Ar-etching. After that, contact pads for the application of the d.c. current, which are made from 3-nm Ti and 50 nm Au, are prepared by evaporation and lift-off. Then, a 70-nm $Al_2O_3$ layer is deposited by atomic layer deposition to electrically isolate the d.c. contacts and the coplanar waveguide (CPW). Finally, the CPW consisting of 5 nm Ti and 150 nm Au is fabricated by evaporation, and the Fe/Al/Pt stripes are located in the gap between the signal line and ground line of the CPW (Fig. 2a). During the fabrication, the highest baking temperature is 110 °C. The CPW is designed to match the radiofrequency network that has an impedance of 50 Ω. The width of the signal line and the gap are 50 µm and 30 µm, respectively. Magnetization dynamics of Fe are excited by out-of-plane Oersted field induced by the radiofrequency microwave currents flowing in the signal and ground lines.

### FMR measurements

The FMR method is used in this study for several reasons: (1) FMR has a higher sensitivity than static magnetization measurements. (2) The FMR method, together with angle and frequency-dependent measurements, is a standard way to quantify the effective magnetization, magnetic anisotropies and Gilbert damping. (3) Damping-like and field-like torques can be determined simultaneously in a single experiment, and thus we can establish a connection between damping-like torque and the modification of magnetic anisotropies. (4) The Joule heating effect, which also alters the magnetic properties of Fe, can be easily excluded from the $I$ dependence of $H_R$.

The FMR spectra are measured optically by time-resolved magneto-optical Kerr microscopy; a pulse train of a Ti:sapphire laser (repetition rate of 80 MHz and pulse width of 150 fs) with a wavelength of 800 nm is phase-locked to a microwave current. A phase shifter is used to adjust the phase between the laser pulse train and microwave, and the phase is kept constant during the measurement. The polar Kerr signal at a certain phase, $V_{Kerr}$, is detected by a lock-in amplifier by phase modulating the microwave current at a frequency of 6.6 kHz. The $V_{Kerr}$ signal is measured by sweeping the external magnetic field, and the magnetic field can be rotated in-plane by 360°. A Keithley 2400 device is used as the d.c. current source for linewidth and resonance field modifications. All measurements are performed at room temperature.

The FMR spectra are well fitted by combining a symmetric ($L_{sym} = \Delta H^2/[4(H - H_R)^2 + \Delta H^2]$) and an anti-symmetric Lorentzian ($L_{a\text{-}sym} = -4\Delta H(H - H_R)/[4(H - H_R)^2 + \Delta H^2]$), $V_{Kerr} = V_{sym}L_{sym} + V_{a\text{-}sym}L_{a\text{-}sym} + V_{offset}$, where $H_R$ is the resonance field, $\Delta H$ is the full width at half maximum, $V_{offset}$ is the offset voltage, and $V_{sym}$ ($V_{a\text{-}sym}$) is the magnitude of the symmetric (anti-symmetric) component of $V_{Kerr}$. It is worth mentioning that, by analysing the position of $H_R$, we have also confirmed that the application of the charge currents does not have a detrimental effect on the magnetic properties of the Fe films (Supplementary Note 2).

### Magnetic anisotropies in Pt/Al/Fe/GaAs multilayers

A typical in-plane magnetic field angle $\varphi_H$ dependence of the resonance field $H_R$ for $t_{Fe} = 1.2$ nm measured at $f = 13$ GHz is shown in Extended Data Fig. 2a. The sample shows typical in-plane uniaxial anisotropy with two-fold symmetry, that is, a magnetically HA for $\varphi_H = -45°$ and 135° ($\langle\bar{1}10\rangle$ orientations) and a magnetically EA for $\varphi_H = 45°$ and 225° ($\langle 110\rangle$ orientations), which originates from the anisotropic bonding at the Fe/GaAs interface[33]. To quantify the magnitude of the anisotropies, we further measure the $f$ dependence of $H_R$ both along the EA and the HA (Extended Data Fig. 2b). Both the angle and frequency dependence of $H_R$ are fitted according to[34,50]

$$\left(\frac{2\pi f}{\gamma}\right)^2 = \mu_0^2 H_1^R H_2^R, \tag{5}$$

with $H_1^R = H^R \cos(\varphi - \varphi_H) + H_K + H_B(3 + \cos 4\varphi)/4 - H_U \sin^2(\varphi - 45°)$ and $H_2^R = H^R \cos(\varphi - \varphi_H) + H_B \cos 4\varphi - H_U \sin 2\varphi$. Here $\gamma (= g\mu_B/\hbar)$ is the gyromagnetic ratio, $g$ is the Landé $g$-factor, $\mu_B$ is the Bohr magneton, $\hbar$ is the reduced Planck constant, $H_K (= M - H_\perp)$ is the effective demagnetization magnetic anisotropy field, including the perpendicular magnetic anisotropy field $H_\perp$, $H_B$ is the biaxial magnetic anisotropy field along the $\langle 100\rangle$ orientations, $H_U$ is the in-plane UMA field along $\langle 110\rangle$ orientations and $\varphi$ is the in-plane angle of magnetization as defined in Extended Data Fig. 1. The magnitude of $\varphi$ is obtained by the equilibrium condition

$$H_R \sin(\varphi - \varphi_H) + (H_B/4)\sin 4\varphi + (H_U/2)\cos 2\varphi = 0. \tag{6}$$

It can be checked that $\varphi = \varphi_H$ holds when **H** is along $\langle 110\rangle$ and $\langle\bar{1}10\rangle$ orientations. From the fits of $H_R$, the magnitude of the magnetic anisotropy fields $H_A$ ($H_A = H_K, H_B, H_U$) for each $t_{Fe}$ is obtained, and their dependences on inverse Fe thickness $t_{Fe}^{-1}$, together with the results obtained from the $AlO_x$/Fe/GaAs samples, are shown in Extended Data Fig. 2c. The results show that the Pt/Al/Fe/GaAs samples have virtually identical magnetic anisotropies as the $AlO_x$/Fe/GaAs samples, and introducing the Pt/Al layer neither enhances the magnetization leading to an increase in $H_K$ nor generates a perpendicular anisotropy leading to a decrease in $H_K$. By comparing the values of $H_K$ and $M$, we confirm that the main contribution to $H_K$ stems from the magnetization due to the demagnetization field. For both sample series, $H_K$ and $H_B$ decrease as $t_{Fe}$ decreases because of the reduction of the magnetization as $t_{Fe}$ decreases, and both of them scale linearly with $t_{Fe}^{-1}$. The intercept (about 2,220 mT) of the $H_K - t_{Fe}^{-1}$ trace corresponds to the saturation magnetization of bulk Fe, and the intercept (around 45 mT) of the $H_B - t_{Fe}^{-1}$ trace corresponds to the biaxial anisotropy of bulk Fe. In contrast to $H_K$ and $H_B$, $H_U$ shows a linear dependence on $t_{Fe}^{-1}$ with a zero intercept, indicative of the interfacial origin of $H_U$.

### Effective mixing conductance in Pt/Al/Fe/GaAs multilayers

Extended Data Fig. 3a,b shows the $\varphi_H$ dependence and $f$ dependence, respectively, of linewidth $\Delta H$ for $t_{Fe} = 1.2$ nm. The magnitude of $\Delta H$ varies strongly with $\varphi_H$ because of the presence of in-plane anisotropy, and the dependencies of $\Delta H$ on $f$ along both EA and HA show linear behaviour. Both the angular and frequency dependence of $\Delta H$ can be well fitted by[51]

$$\Delta H = \Delta[\mathrm{Im}(\chi)] + \Delta H_0$$
$$= \Delta\left[\frac{\alpha\sqrt{H_1^R H_2^R}(H_1 H_1 + H_1^R H_2^R)M}{(H_1 H_2 - H_1^R H_2^R)^2 + \alpha^2 H_1^R H_2^R(H_1 + H_2)^2}\right] + \Delta H_0, \tag{7}$$

where $\Delta[\mathrm{Im}(\chi)]$ is the linewidth of the imaginary part of the dynamic magnetic susceptibility $\mathrm{Im}(\chi)$, $H_1$ and $H_2$ are defined in equation (5) for arbitrary $H$ values, and $\Delta H_0$ is the residual linewidth (zero-frequency intercept). As the angular trace can be well fitted by using a damping value of 0.0078, there is no need to consider other extrinsic effects (that is, inhomogeneity and/or two-magnon scattering) contributing to $\Delta H$. It is worth mentioning that the angular trace gives a slightly higher $\alpha$ value because $\Delta H_0$, which also depends on $\varphi_H$, is not considered in the fit. In this case, the frequency dependence of linewidth gives more reliable damping values (Extended Data Fig. 3b). Extended Data Fig. 3c compares the magnitude of damping for Pt/Al/Fe/GaAs and AlO$_x$/Fe/GaAs samples. For both sample series, the Gilbert damping increases as $t_{\mathrm{Fe}}$ decreases and a linear dependence of $\alpha$ on $t_{\mathrm{Fe}}^{-1}$ is observed. The enhancement of $\alpha$ is because of the spin pumping effect, which is given by[52,53]

$$\alpha = \alpha_0 + g_{\mathrm{eff}}^{\uparrow\downarrow}\frac{\gamma\hbar}{4\pi M}t_{\mathrm{Fe}}^{-1}, \tag{8}$$

where $\alpha_0$ is the intrinsic damping of pure bulk Fe and $g_{\mathrm{eff}}^{\uparrow\downarrow}$ is the effective spin mixing conductance quantifying the spin pumping efficiency. By using $\mu_0 M = 2.2$ T and $\gamma = 1.80 \times 10^{11}$ rad s$^{-1}$ T$^{-1}$, the magnitude of $g_{\mathrm{eff}}^{\uparrow\downarrow}$ for Pt/Al/Fe/GaAs is determined to be $4.6 \times 10^{18}$ m$^{-2}$, and $g_{\mathrm{eff}}^{\uparrow\downarrow}$ at the Fe/GaAs interface is determined to be $1.9 \times 10^{18}$ m$^{-2}$. Therefore, by subtracting these two values, the magnitude of $g_{\mathrm{eff}}^{\uparrow\downarrow}$ at Pt/Al/Fe interface is determined to be $2.7 \times 10^{18}$ m$^{-2}$. The spin transparency $T_{\mathrm{int}}$ of the Pt/Al/Fe interface is given by ref. 53

$$T_{\mathrm{int}} = \frac{2e^2}{h}\frac{g_{\mathrm{eff}}^{\uparrow\downarrow}}{G_{\mathrm{Pt}}} \tag{9}$$

where $2e^2/h$ is the conductance quantum, $G_{\mathrm{Pt}} [= 1/(\rho_{xx}\lambda_s)]$ is the spin conductance of Pt, $\rho_{xx}$ is the resistivity and $\lambda_s$ is the spin diffusion length. By using $\lambda_s = 4$ nm and an averaged $\rho_{xx} = 40$ $\mu\Omega$ cm, $T_{\mathrm{int}} = 0.21$ is determined. We note that the magnitude of $g_{\mathrm{eff}}^{\uparrow\downarrow}$ at the Pt/Al/Fe interface is about one order of magnitude smaller than the experimental values found at heavy metal/ultrathin ferromagnet interfaces[54], but very close to the value obtained by the first-principles calculations[55]. The previously overestimated $g_{\mathrm{eff}}^{\uparrow\downarrow}$ and thus $T_{\mathrm{int}}$ at heavy metal/ultrathin ferromagnet interfaces is probably because the enhancement of $\alpha$ by two-magnon scattering[56] as well as by the magnetic proximity effect (see Supplementary Note 3) is not properly excluded. Moreover, the obtained $\alpha_0$ values for Pt/Al/Fe/GaAs ($\alpha_0 = 0.0039$) and AlO$_x$/Fe/GaAs ($\alpha_0 = 0.0033$) slightly differ; the reason is unclear to us, but might be because of a small error in the Fe thickness, which is hard to be determined accurately in the ultrathin regime.

### Theory of the modulation of the linewidth

To model the modulation of the FMR linewidth by the application of d.c. current, the Landau–Lifshitz–Gilbert equation with damping-like spin-torque term is considered[18,35],

$$\frac{d\mathbf{M}}{dt} = -\gamma\mathbf{M}\times\mu_0\mathbf{H}_{\mathrm{eff}} + \frac{\alpha}{M}\mathbf{M}\times\frac{d\mathbf{M}}{dt} - \frac{\gamma\mu_0 h_{\mathrm{DL}}}{M}\mathbf{M}\times\mathbf{M}\times\boldsymbol{\sigma}. \tag{10}$$

The terms on the right side of equation (10) correspond to the precession torque, the damping torque and the damping-like spin torque induced by the spin current. Here $\boldsymbol{\sigma}$ is the spin polarization unit vector, and $h_{\mathrm{DL}}$ is the effective anti-damping-like magnetic field. The effective magnetic field $\mathbf{H}_{\mathrm{eff}}$, containing both external and internal fields, is expressed in terms of the free energy density $F$, which can be obtained as

$$\mathbf{H}_{\mathrm{eff}} = -\frac{1}{\mu_0}\frac{\partial F}{\partial\mathbf{M}}. \tag{11}$$

For single-crystalline Fe films grown on GaAs(001) substrates with in-plane magnetic anisotropies, $F$ is given by[34,58]

$$F = \frac{\mu_0 M}{2}\left\{-2H[\cos\theta\cos\theta_H + \sin\theta\sin\theta_H\cos(\varphi - \varphi_H)] + H_K\cos^2\theta\right.$$
$$\left. - \frac{H_B}{2}\sin^4\theta\frac{3+\cos4\varphi}{4} - H_U\sin^2\theta\sin^2\left(\varphi - \frac{\pi}{4}\right)\right\}. \tag{12}$$

Bringing equations (11) and (12) into equation (10), the time-resolved magnetization dynamics for current flowing along the [110] orientation (that is, $\boldsymbol{\sigma}\parallel[\bar{1}10]$) is obtained as

$$\begin{cases} \dfrac{\partial\varphi}{\partial t} = \dfrac{\gamma\mu_0}{(1+\alpha^2)M\sin\theta}\left(\dfrac{\partial F}{\partial\theta} - \dfrac{\alpha}{\sin\theta}\dfrac{\partial F}{\partial\varphi}\right) \\ \qquad\quad + \dfrac{\gamma\mu_0 h_{DL}}{(1+\alpha^2)\sin\theta}\dfrac{\sqrt{2}}{2}[\alpha\cos\theta(\sin\varphi - \cos\varphi) + \cos\varphi \\ \qquad\quad + \sin\varphi] \\ \dfrac{\partial\theta}{\partial t} = \dfrac{\gamma\mu_0}{M\sin\theta}\left(\dfrac{\alpha^2}{1+\alpha^2} - 1\right)\dfrac{\partial F}{\partial\varphi} - \dfrac{\alpha}{1+\alpha^2}\dfrac{\gamma\mu_0}{M}\dfrac{\partial F}{\partial\theta} \\ \qquad\quad + \left(1 + \dfrac{\alpha^2}{1+\alpha^2}\right)\gamma\mu_0 h_{DL}\dfrac{\sqrt{2}}{2}\cos\theta(\sin\varphi - \cos\varphi) \\ \qquad\quad + \dfrac{\alpha}{1+\alpha^2}\gamma\mu_0 h_{DL}\dfrac{\sqrt{2}}{2}(\cos\varphi + \sin\varphi) \end{cases} \tag{13}$$

Similarly, for the current flowing along the [100]-orientation (that is, $\boldsymbol{\sigma}\parallel[010]$), we have

$$\begin{cases} \dfrac{\partial\varphi}{\partial t} = \dfrac{\gamma\mu_0}{(1+\alpha^2)M\sin\theta}\left(\dfrac{\partial F}{\partial\theta} - \dfrac{\alpha}{\sin\theta}\dfrac{\partial F}{\partial\varphi}\right) \\ \qquad\quad + \dfrac{\gamma\mu_0 h_{DL}}{(1+\alpha^2)\sin\theta}(\alpha\cos\theta\sin\varphi + \cos\varphi) \\ \dfrac{\partial\theta}{\partial t} = \dfrac{\gamma\mu_0}{M\sin\theta}\left(\dfrac{\alpha^2}{1+\alpha^2} - 1\right)\dfrac{\partial F}{\partial\varphi} - \dfrac{\alpha}{1+\alpha^2}\dfrac{\gamma\mu_0}{M}\dfrac{\partial F}{\partial\theta} \\ \qquad\quad - \gamma\mu_0 h_{DL}\left[\dfrac{\alpha^2}{1+\alpha^2}(\alpha\cos\theta\sin\varphi + \cos\varphi) - \cos\theta\sin\varphi\right] \end{cases} . \tag{14}$$

The time dependence of $\varphi(t)$, $\theta(t)$ and then $\mathbf{m}(t)$ can be readily obtained from equations (13) and (14), and Extended Data Fig. 4a shows an example of the time-dependent $m_z$ by using $\mu_0 H = 101$ mT, $\mu_0 H_K = 1{,}350$ mT, $\mu_0 H_U = 128$ mT, $\mu_0 H_B = 10$ mT, $\alpha = 0.0063$ and $\mu_0 H_{DL} = 0$. The damped oscillating dynamic magnetization can be well fitted by

$$m_z(t) = A e^{-t/\tau}\cos(2\pi ft + \phi) \tag{15}$$

where $A$ is the amplitude, $\tau$ is the magnetization relaxation time and $\phi$ is the phase shift. The connection between $\tau$ and $\Delta H$ is given by

$$\Delta H = \frac{1}{2\pi}\left|\frac{dH_R}{df}\right|\frac{1}{\tau} \tag{16}$$

where $dH_R/df$ can be readily obtained from equation (5). We confirm the validity of the above method in Extended Data Fig. 4b by showing that the angle dependence of $\Delta H$ obtained from the time domain (equation (16)) at $h_{DL} = 0$ is identical to the linewidth obtained by the dynamic susceptibility in the magnetic field domain (equation (7)).

Having obtained the linewidth for $I = 0$, the next step is to calculate the influence of the linewidth by spin–orbit torque. The magnitude of $h_{DL}$ is given by

$$\mu_0 h_{DL} = \frac{\hbar}{2e} \frac{\xi}{M t_{Fe}} j_{Pt} \quad (17)$$

where $\xi$ is the effective damping-like torque efficiency and $j_{Pt}$ is the current density in Pt. For the Pt/Al/Fe multilayer, $j_{Pt}$ is determined by the parallel resistor model

$$j_{Pt} = \frac{t_{Pt}\rho_{Al}\rho_{Fe}}{t_{Pt}\rho_{Al}\rho_{Fe} + t_{Al}\rho_{Pt}\rho_{Fe} + t_{Fe}\rho_{Pt}\rho_{Al}} \frac{I}{w t_{Pt}} \quad (18)$$

where $\rho_{Pt}$ (= 40 μΩ cm), $\rho_{Al}$ (= 10 μΩ cm) and $\rho_{Fe}$ (= 50 μΩ cm) are the resistivities of the Pt, Al and Fe layers, respectively; $t_{Pt}$, $t_{Al}$ and $t_{Fe}$ are the thicknesses of the Pt, Al and Fe layers, respectively; $I$ is the d.c. current; and $w$ is the width of the device. Plugging equations (17) and (18) into equations (13) and (14), the $I$ dependence of $\Delta H$ can be obtained. An example is shown in Extended Data Fig. 4c, which shows a linear $\Delta H$–$I$ relationship. From the linear fit (equation (1) in the main text), we obtain the modulation amplitude of $\Delta H$, that is, $d(\Delta H)/dI$. Extended Data Fig. 4d presents the calculated $d(\Delta H)/dI$ as a function of the magnetic field angle, which shows a strong variation around the HA.

To reproduce the experimental data as shown in Fig. 1f in the main text, the magnitude of the magnetic anisotropies and the damping parameter obtained in Extended Data Fig. 3 as well as $\xi = 0.06$ are used. Note that the distinctive presence of robust UMA at the Fe/GaAs interface significantly alters the angular dependence of $d(\Delta H)/dI$. This deviation is remarkable when compared with the $\sin\varphi_{I-H}$ dependence of $d(\Delta H)/dI$ as observed in polycrystalline samples, such as Pt/Py (refs. 57,58).

To understand the strong deviation of $d(\Delta H)/dI$ around the HA, we plot the in-plane angular dependence of $F$ in Extended Data Fig. 5 for $\theta = \theta_H = 90°$, that is,

$$F = \frac{\mu_0 M}{2}\left[ -2H_R\cos(\varphi - \varphi_H) - \frac{H_B}{2}\frac{3 + \cos4\varphi}{4} - H_U\sin^2\left(\varphi - \frac{\pi}{4}\right)\right]. \quad (19)$$

It shows that, around the HA (approximately ±15°), the magnetic potential barrier completely vanishes and $\frac{\partial F}{\partial\varphi} = 0$ and $\frac{\partial^2 F}{\partial\varphi^2} < 0$ hold. This indicates that the net static torques induced by internal and external magnetic fields acting on the magnetization cancel and the magnetization has a large cone angle for precession[59]. Consequently, the magnetization behaves freely with no constraints in the vicinity of the HA, and the low stiffness allows larger $d(\Delta H)/dI$ values induced by spin current[60]. If there are no in-plane magnetic anisotropies, the free energy is constant and is independent of the angle, the magnetization always follows the direction of the applied magnetic field and has the same stiffness at each position. Therefore, the modulation shows no deviation around the HA.

## Frequency dependence of the linewidth modulation

Extended Data Fig. 6a shows the frequency dependence of the modulation of linewidth $d(\Delta H)/dI$ for $t_{Fe} = 2.8$ nm and 1.2 nm, in which the current flows along the [100] orientation. For both samples, the modulation changes polarity as the direction of **M** is changed by 180°. The modulation amplitude increases quasi-linearly with frequency, and the experimental results can be also reproduced by equation (14) using $\xi = 0.06$, consistent with the angular modulation shown in Fig. 2f. For **H** along the ⟨110⟩ and ⟨Ī10⟩ orientations, the frequency and the Fe thickness dependence of linewidth modulation is approximately given by[24]

$$\frac{d(\mu_0\Delta H)}{d(I)} = 2\frac{2\pi f}{\gamma}\frac{\sin\varphi_{I-H}}{H_R + H_K/2}\frac{\hbar}{2e}\frac{\xi}{M t_{Fe}}\frac{1}{t_{Pt}w}, \quad (20)$$

where $\varphi_{I-H} = 45°, 135°, 225°$ and $315°$ as shown by the inset of each panel in Extended Data Fig. 6. The damping-like torque efficiency can be

further quantified by the slope $s$ of $f$-dependence modulation, that is, $s = \frac{d[d(\Delta H)/dI]}{df}$. Extended Data Fig. 7 shows the absolute value of $s$ values as a function of $t_{Fe}^{-1}$. A linear dependence of $|s|$ on $t_{Fe}^{-1}$ is observed, which indicates that the damping-like torque is an interfacial effect, originating from the absorption of spin current generated in Pt (ref. 61).

## Quantifying the modification of the magnetic anisotropies

In this section, we show our procedure to quantify the modulation of magnetic anisotropies by spin currents. According to equation (5), the $f$ dependencies of $H_R$ along the EA ($\varphi_H = \varphi = 45°$ and 225°) and the HA ($\varphi_H = \varphi = 135°$ and 315°) are given by equation (3). From the angle and frequency dependencies of $H_R$ as shown in Extended Data Fig. 2, $\mu_0 H_K = 1{,}350$ mT, $\mu_0 H_U = 128$ mT, $\mu_0 H_B = 10$ mT and $g = 2.05$ are determined for $t_{Fe} = 1.2$ nm. Extended Data Fig. 8a shows the $H_R$ dependence of $f$ for $\mu_0 H_K = 1{,}350$ mT (blue solid line) and $\mu_0 H_K + \Delta\mu_0 H_K = 1{,}400$ mT (red solid line) along the HA calculated by equation (3). To exaggerate the difference, $\mu_0 H_K$ of 50 mT is assumed. The shift of the resonance field $\Delta H_R$ is obtained as $\Delta H_R = H_R(H_K) - H_R(H_K + \Delta H_K)$, and the frequency dependence of $\Delta H_R$ is plotted in Extended Data Fig. 8b, which shows a linear behaviour with respect to $f$ between 10 GHz and 20 GHz (in the experimental range), that is, $\Delta H_R = k_K f$. Note that, to simplify the analysis, the zero-frequency intercept is ignored because the magnitude is much smaller than the intercept induced by $\Delta H_U$ and $\Delta H_B$. The sign of the slope $k_K$ is the same as that of $\Delta H_K$ and its magnitude is proportional to $\Delta H_K$, that is, $k_K \propto \Delta H_K$. For the EA as shown in Extended Data Fig. 8c,d, the $\Delta H_R$–$f$ relationship induced by $\Delta H_K$ remains the same as for the HA, that is, $\Delta H_R = k_K f$ still holds.

Extended Data Fig. 8e shows the $H_R$ dependence of $f$ for $\mu_0 H_U = 128$ mT (blue solid line) and $\mu_0 H_U + \mu_0\Delta H_U = 178$ mT (red solid line) along the HA. As shown in Extended Data Fig. 8f, the shift of the resonance field along the HA is independent of $f$ with a negative intercept, that is, $\Delta H_R = -\Delta H_U$. However, for the EA, as shown in Extended Data Fig. 8g,h, the $f$-dependent $\Delta H_R$ can be expressed as $\Delta H_R = \Delta H_U - k_U f$, which has an opposite slope compared with the $\Delta H_R$–$f$ relationships induced by $\Delta H_K$ (Extended Data Fig. 8d), that is, $k_U \propto -\Delta H_U$.

If the modulation is induced by a change in the biaxial anisotropy as shown in Extended Data Fig. 8i–l, $\Delta H_R$ along both the HA and EA shows a linear dependence on $f$, which is expressed as $\Delta H_R = -\Delta H_B + k_B f$, and $k_B \propto \Delta H_B$ holds.

Extended Data Table 1 summarizes the $\Delta H_R$–$f$ relationships both along the EA and HA induced by $\Delta H_K$, $\Delta H_U$ and $\Delta H_B$.

As $h_{Oe/FL}$ generated by the d.c. current also shifts the resonance field along the EA and HA axes by $\pm\frac{\sqrt{2}}{2}h_{Oe/FL}$, where plus corresponds to the [110] (EA) and the [Ī10] (HA) directions, and minus corresponds to the [Ī10] (EA) and the [1Ī0] (HA) directions, the total $\Delta H_R$ induced by $\Delta H_K$, $\Delta H_U$ and $\Delta H_B$ along the EA and HA is, respectively, given by equation (4).

Based on equations (4) and (5), the values of $\Delta H_K$, $\Delta H_U$, $\Delta H_B$ and $h_{Oe/FL}$ for $t_{Fe} \leq 2.2$ nm are extracted as follows:

1. We consider the results obtained for **H** ∥ **M** ∥ [110] (EA) and **H** ∥ **M**/[Ī10] (HA) as shown in Extended Data Fig. 9a (the same results as shown in Fig. 4 in the main text for $I = 1$ mA), where the net magnetization is parallel to **I**. At $f = 0$, equation (4) is reduced to

$$\Delta H_R^{EA}(0) = \Delta H_U - \Delta H_B + \frac{\sqrt{2}}{2}h_{Oe/FL} = -0.20 \text{ mT} \quad (21)$$

$$\Delta H_R^{HA}(0) = -(\Delta H_U + \Delta H_B) - \frac{\sqrt{2}}{2}h_{Oe/FL} = -0.32 \text{ mT}. \quad (22)$$

By adding equations (21) and (22), the magnitude of $\Delta H_B$ is determined to be 0.26 mT, which corresponds to $k_B$ of $4 \times 10^{-3}$ mT GHz$^{-1}$ according to equation (3).

2. From Extended Data Fig. 9a, the slope along the HA is determined to be $k_K + k_B = 0.025$ mT GHz$^{-1}$. Thus, the magnitude of $k_K$ is determined

by $k_K = 0.025$ mT GHz$^{-1}$ − $k_B = 0.021$ mT GHz$^{-1}$, which corresponds to $\Delta H_K = 2.0$ mT according to equation (3).

3. As $\Delta H_R^{EA}$ is frequency independent, this requires that $k_U = k_K + k_B = 0.025$ mT GHz$^{-1}$, which corresponds $\Delta H_U = 2.5$ mT.

4. As the magnetization along EA and HA is, respectively, rotated by 180° to the $[\bar{1}\bar{1}0]$ and $[\bar{1}10]$ directions, and the net magnetization is antiparallel to **I** (Extended Data Fig. 9b), we obtain $\Delta H_B = -0.26$ mT, $\Delta H_K = -2.0$ mT and $\Delta H_U = -2.5$ mT, which are of opposite sign as the results obtained from Extended Data Fig. 9a.

5. Finally, bringing the magnitude of $\Delta H_B$ and $\Delta H_U$ back into equations (21) and (22), $\frac{\sqrt{2}}{2}h_{Oe/FL}$ is determined to be −2.24 mT. The negative sign of $h_{Oe/FL}$ indicates that it is along the $[0\bar{1}0]$ orientation.

Similarly, the corresponding $\Delta H_B$, $\Delta H_K$ and $\Delta H_U$ values can be determined for $t_{Fe} = 2.2$ nm (Extended Data Fig. 10). Extended Data Table 2 summarizes the magnitudes of the magnetic anisotropy modifications as well as the $h_{Oe/FL}$ values for all the devices. The enhancement of the field-like torque in thinner samples has been observed in other systems and is probably because of the enhanced Bychkov–Rashba spin–orbit interaction[61,62] and/or the orbital angular momentum (orbital Hall effect and orbital Rashba effect) at the ferromagnetic metal/heavy metal interface[62].

It is worth mentioning that, once the magnetization direction is fixed, $\Delta H_B$, $\Delta H_K$ and $\Delta H_U$ obtained either from Extended Data Fig. 9a (Extended Data Fig. 10a) or from Extended Data Fig. 9b (Extended Data Fig. 10b) have the same sign (either positive or negative depending on the direction of **M**). This is consistent with the change in magnetic anisotropies by temperature (Supplementary Fig. 7), which shows that the magnitude of $\Delta H_B$, $H_K$ and $\Delta H_U$ increases as the temperature decreases and decreases as the temperature increases. This indicates that the increase in the magnetic anisotropies is dominated by the increase in $M$ as temperature decreases and the decrease in the magnetic anisotropies is dominated by the decrease in $M$ as temperature increases. For the spin current modification demonstrated here, the temperature is not changed but the change in $M$ is induced by populating the electronic bands by the spin current. More interestingly, the new modification method can control the increase or decrease in $M$ simply by the direction of current and/or the direction of magnetization, which is not accessible by other controls.

### Alternative interpretation of the experimental results

It is known that the starting point of the FMR analysis is the static magnetic energy landscape, which is related to the magnetic anisotropies. Therefore, it is natural to consider that the modification of magnetic anisotropy accounts for the $f$-linear $dH_R/dI$ curves as observed in the experiment. Although the data analysis discussed in the previous section is self-consistent, there could be alternative interpretations of the data. One possibility could be the current-induced modification of the Landé $g$-factor of Fe. In magnetic materials, it is known that $g$ is related to the orbital moment $\mu_L$ and the spin moment $\mu_S$:

$$g = \frac{2\mu_L}{\mu_S} + 2. \tag{23}$$

A flow of spin and orbital angular momentum induced by charge current could, respectively, modify the orbital and spin moment of Fe by $\Delta\mu_S$ and $\Delta\mu_L$, and then a change in the gyromagnetic ratio of Fe is expected. This could, in turn, lead to a shift of FMR resonance fields linearly depending on the frequency. However, if this were the case, an anisotropic modification of $g$ is needed to interpret the data as observed in Extended Data Figs. 9 and 10 (that is, there is sizeable modification along the HA, but no modification along the EA). As we cannot figure out why the modification of $g$ could be anisotropic, we ignore the discussion of the $g$-factor modification in the main text.

We are also open to other possible explanations for the experimental observations.

### Estimation of the magnitude of spin transfer electrons

The change in magnetization is attributed to the additional filling of the electronic $d$-band. The induced filling of the bands in Fe occurs mainly close to the interface and is not homogeneously distributed, as it depends on the spin diffusion length of the spin current in Fe. In other words, the measured modulated magnetic anisotropies are averaged over the whole ferromagnetic film. For simplicity, we neglect the spin current distribution in Fe and assume that it is homogeneously distributed. The spin chemical potential at the interface[63] is given by $u_s^0 = 2e\lambda\xi E \tanh\left(\frac{t_{Pt}}{2\lambda}\right)$, where $e$ is the elementary charge, $\lambda$ is the spin diffusion length, $E (= j/\sigma)$ is the electric field, $j$ is the current density and $\sigma$ is the conductivity of Pt. The areal spin density $n_s$ transferred into Fe is obtained as $n_s = u_s^0\lambda N$ (ref. 18), where $N$ is the density of states at the Fermi level. Using $N = 6 \times 10^{48}$ J$^{-1}$ m$^{-3}$, $\lambda = 4$ nm, $\xi = 0.06$, $\sigma = 2.0 \times 10^6$ Ω$^{-1}$ m$^{-1}$, $n_s = 4.2 \times 10^{12}$ $\mu_B$ cm$^{-2}$ is obtained for $I = 1$ mA. As Fe has a bcc structure (lattice constant $a = 2.8$ Å) with a moment of about 1.0 $\mu_B$ for $t_{Fe} = 1.2$ nm at room temperature[64], the areal density of the magnetic moment of Fe $n_{Fe}$ is determined to be $2.6 \times 10^{14}$ $\mu_B$ cm$^{-2}$. In this case, the filling of the $d$-band by spin current leads to a change in the magnetic moment of the order of $n_s/n_{Fe} \approx 0.16\%$, which agrees with the ratio between $\Delta H_K$ and $H_K$, that is, $\Delta H_K/H_K \approx 2.0$ mT/ 1 T $\approx 0.2\%$.

### Data availability

The experimental and theoretical calculation data used in this paper are freely available at the open science framework https://doi.org/10.17605/OSF.IO/RZMUJ. Source data are provided with this paper.

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

**Acknowledgements** We thank S. S. Fülöp, J. Shao and J. P. Guo for technical help and M. Stiles for the discussions. This work was funded by the Deutsche Forschungsgemeinschaft by TRR 360–492547816 and SFB1277-314695032, by the excellence cluster MCQST under Germany's Excellence Strategy EXC-2111 (Project no. 390814868), and by FLAG-ERA JTC 2021-2DSOTECH.

**Author contributions** L.C. planned the study. Y.S. and L.C. fabricated the devices and collected the data. L.C. analysed the data and did the theoretical calculations. T.N.G.M. and M.K. grew

the samples. C.S. and A.O. did the high-resolution transmission electron microscopy measurements. S.M. and H.E. did the first-principles calculations. L.C. wrote the paper with input from all other co-authors. All authors discussed the results and contributed to the paper.

**Funding** Open access funding provided by Technische Universität München.

**Competing interests** The authors declare no competing interests.

**Additional information**
**Correspondence and requests for materials** should be addressed to L. Chen.

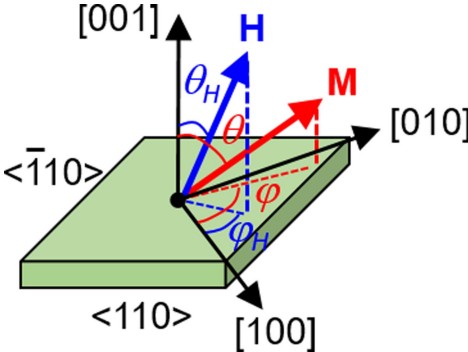

**Extended Data Fig. 1 | Schematic of the coordinate system used for the analysis.** $\theta_H$ and $\varphi_H$ represent the polar and azimuthal angles of external magnetic-field **H**, and $\theta$ and $\varphi$ are the polar and azimuthal angles of magnetization **M**. The Fe/GaAs thin films show competing in-plane magnetic anisotropies along <100>, <110> and $\langle \bar{1}10 \rangle$-orientations.

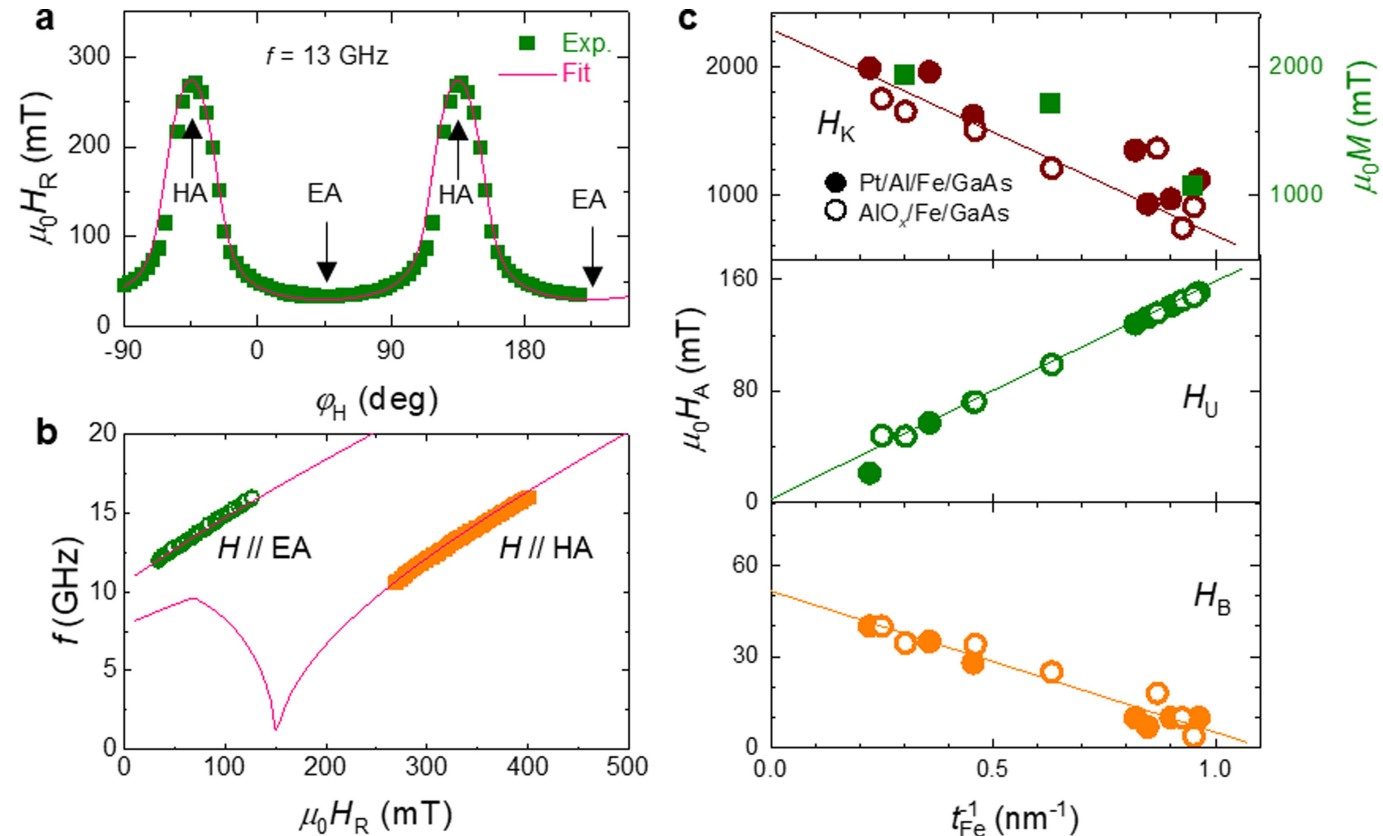

**Extended Data Fig. 2 | Magnetic anisotropies of Fe/GaAs(001).**
**a**, $\varphi_H$-dependence of the resonance field $H_R$ measured for $t_{Fe}$ = 1.2 nm at $f$ = 13 GHz.
**b**, $H_R$-dependence of $f$ measured along the hard axis (HA) and easy axis (EA).
In **a** and **b**, the symbols are the experimental data, and the solid lines are the fits by Eq. (5). **c**, Inverse Fe thickness $t_{Fe}^{-1}$ dependence of $H_K$ (circles) as well as $M$ (squares), $H_U$, and $H_B$ for Pt/Al/Fe/GaAs (solid circles) and AlO$_x$/Fe/GaAs (open circles). The solids lines are the linear fits.

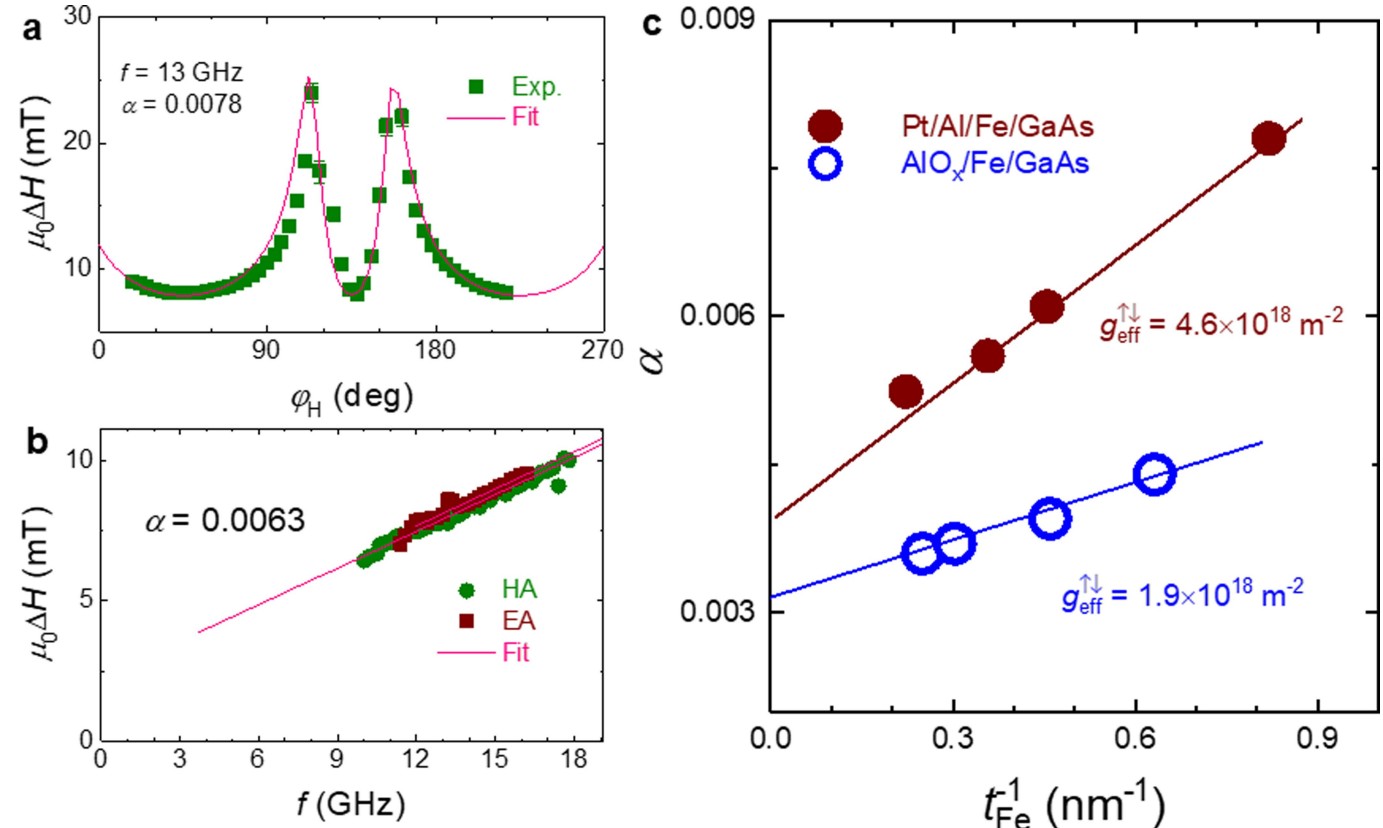

**Extended Data Fig. 3 | Damping and mixing conductance of Fe/GaAs(001). a**, $\varphi_H$-dependence of $\Delta H$ for $t_{Fe}$ = 1.2 nm measured at $f$ = 13 GHz. The solid line is fitted using a damping value of 0.0078. **b**, $f$-dependence of $\Delta H$ measured along the EA and HA. The solid lines are the fits by a damping value of 0.0063.

**c**, $t_{Fe}^{-1}$-dependence of $\alpha$ for Pt/Al/Fe/GaAs samples (solid circles) as well as AlO$_x$/Fe/GaAs samples (open circles). The solid lines are the fits according to spin pumping.

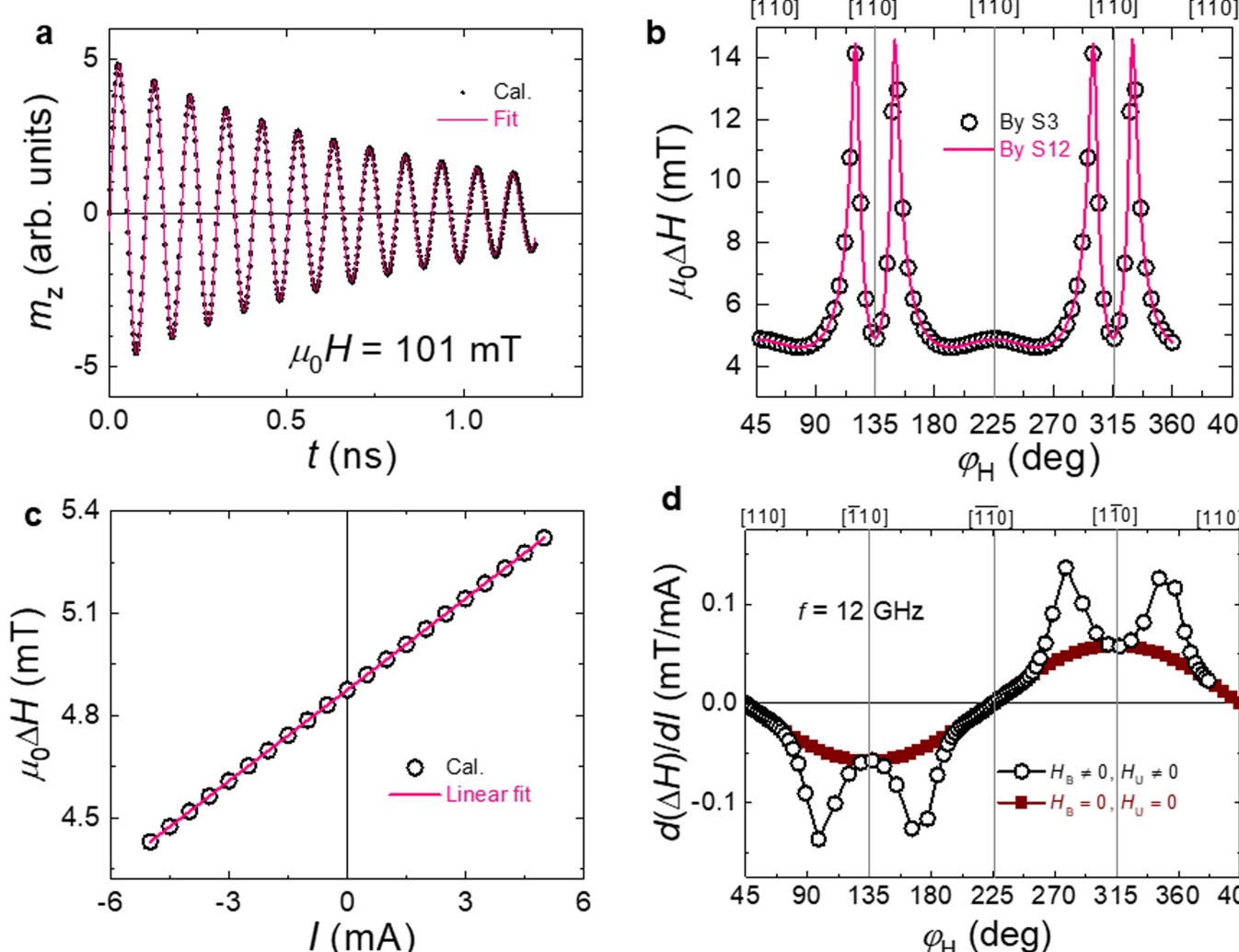

**Extended Data Fig. 4 | Calculation of the linewidth modulation by LLG equation with conventional SOT term. a**, Time-resolved dynamic magnetization calculated by Eq. (13) for $\mu_0 H = 101$ mT. By fitting the damped oscillation of the dynamic magnetization (solid line) by Eq. (15), the magnetization relaxation time is obtained. **b**, Calculated $\varphi_H$-dependence of $\Delta H$ by Eq. (7) and Eq. (16) using $\alpha = 0.0063$. Both methods show identical results. **c**, Calculated $I$-dependence of $\Delta H$; the solid line is the linear fit from which $d(\Delta H)/dI$ is obtained. **d**, Comparison of the $\varphi_H$-dependence of $d(\Delta H)/dI$ calculated with in-plane anisotropy (open circles) and without in-plane anisotropies (solid squares).

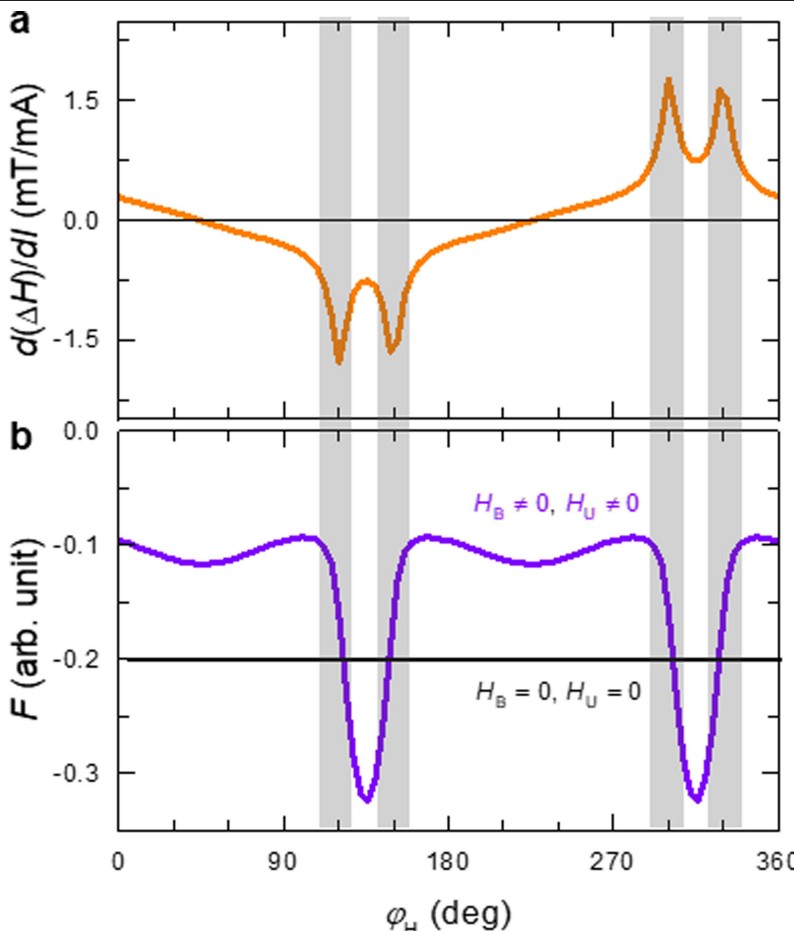

**Extended Data Fig. 5 | Angular dependence of linewidth modification and free energy. a**, $\varphi_H$-dependence of the calculated modulation of linewidth $d(\Delta H)/dI$. **b**, $\varphi_H$-dependence of free energy $F$. Around the HA (shaded areas), the energy barrier vanishes and all the static torques acting on **M** cancel. In this case, the magnetization has a larger precessional cone angle, leading to an enhanced $d(\Delta H)/dI$ values.

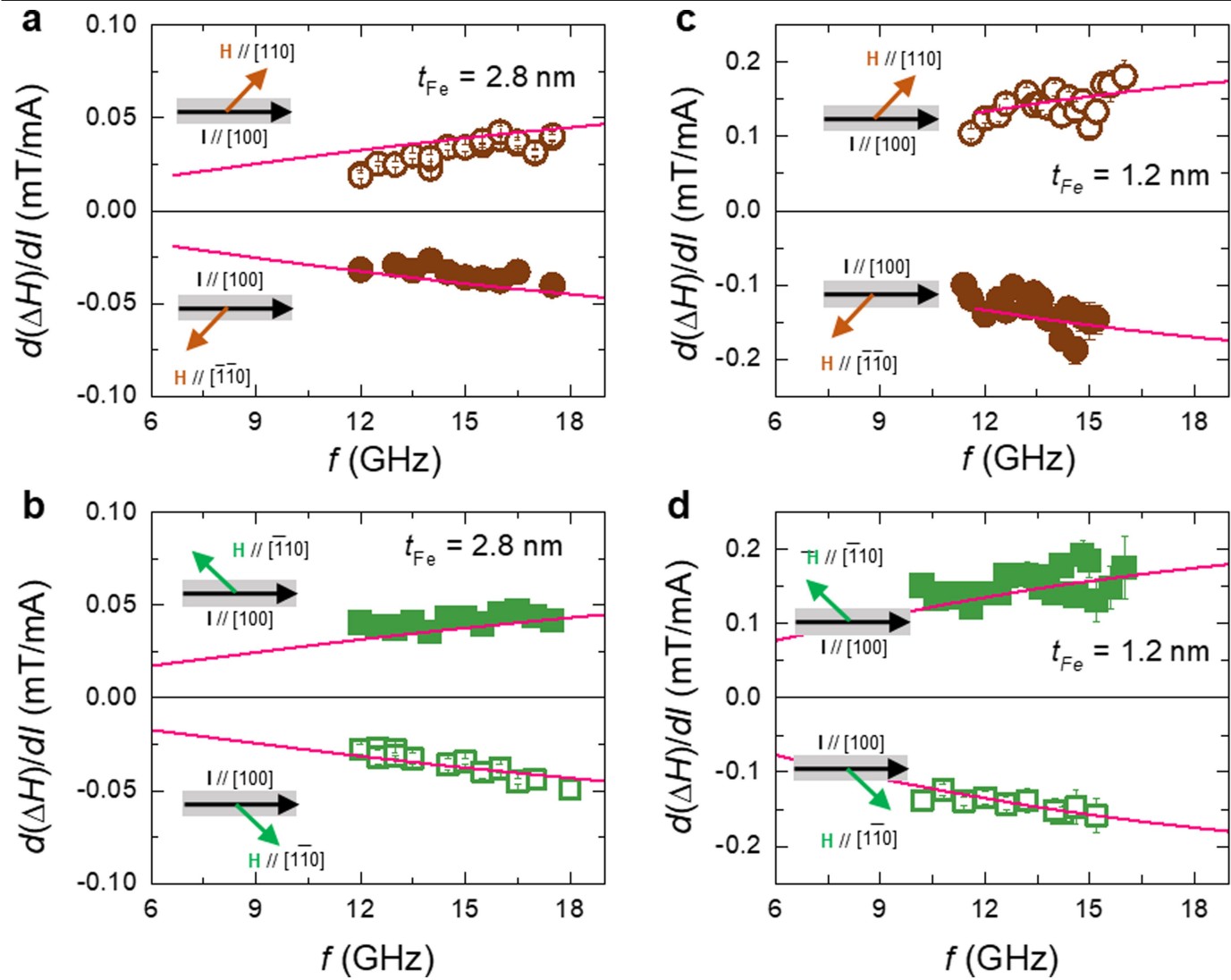

**Extended Data Fig. 6 | Frequency dependence of linewidth modification.** **a**, Frequency dependence of $d(\Delta H)/dI$ for $H$ along the easy axis ([110]- and [$\bar{1}\bar{1}$0] -orientations). **b**, Frequency dependence of $d(\Delta H)/dI$ for $H$ along the hard axis ([$\bar{1}$10]- and [1$\bar{1}$0]-orientations). The results in **a** and **b** are obtained for $t_{Fe}$ = 2.8 nm. **c** and **d** are the same results as **a** and **b** but for $t_{Fe}$ = 1.2 nm. The inset of each figures shows the respective orientation of the charge current and magnetic-field (magnetization). The solid lines in each panel are calculated by Eq. (14) using $\xi$ = 0.06.

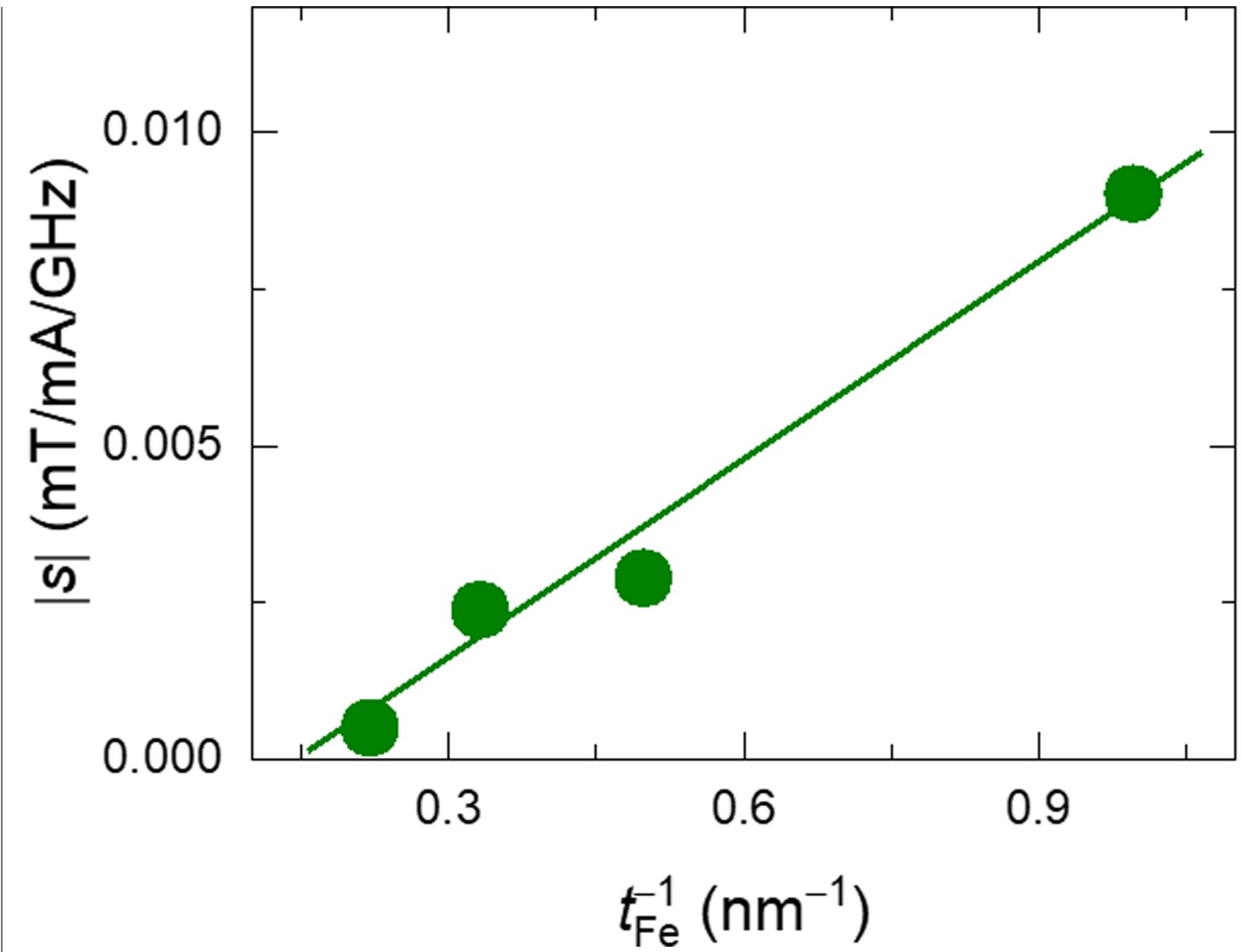

**Extended Data Fig. 7 | dependence of damping-like SOT.** $t_{Fe}^{-1} t_{Fe}^{-1}$ dependence of $|s|$ extracted from Extended Data Fig. 6, where $s = \frac{d[d(\Delta H)/dI]}{df}$. The linear dependence indicates that the damping-like SOT is an interfacial behavior.

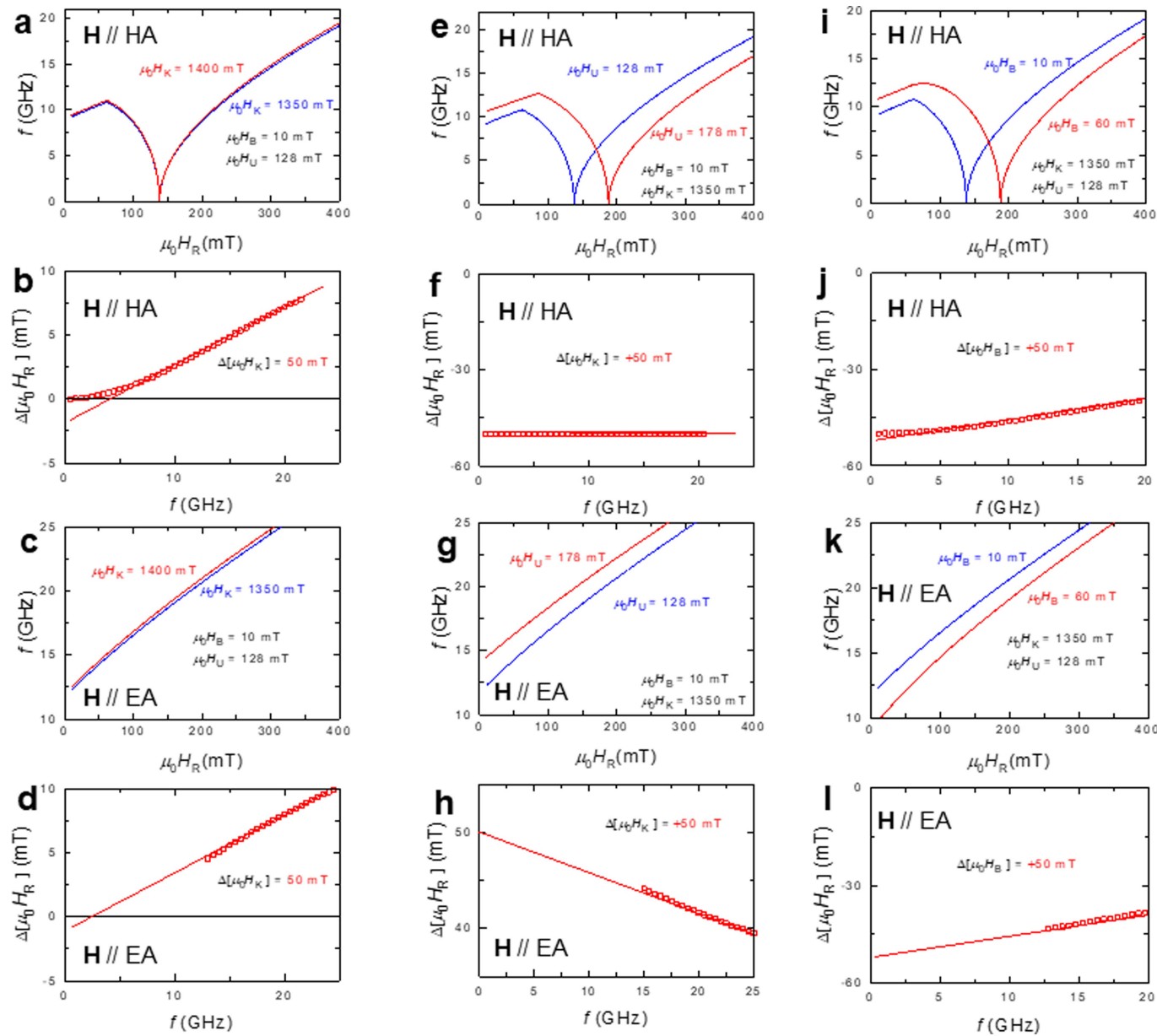

**Extended Data Fig. 8 | Shift of resonance field by magnetic anisotropies.**
**a**, $H_R$-dependence of $f$ calculated for $\mu_0 H_K = 1350$ mT (blue) and $\mu_0 H_K + \mu_0 \Delta H_K = 1400$ mT (red) along the hard axis. **b**, Shift of the resonance field $\Delta H_R$ as a function of frequency, where $\Delta H_R = H_R(H_K) - H_R(H_K + \Delta H_K)$. **c** and **d** are the same results as those in **a** and **b** but for the calculation along the easy axis. **e**-**h** for $\Delta H_U$. **i**-**l** for $\Delta H_B$. In the calculation, a change of magnetic anisotropy fields of 50 mT is assumed for each case to exaggerate the shift of $H_R$.

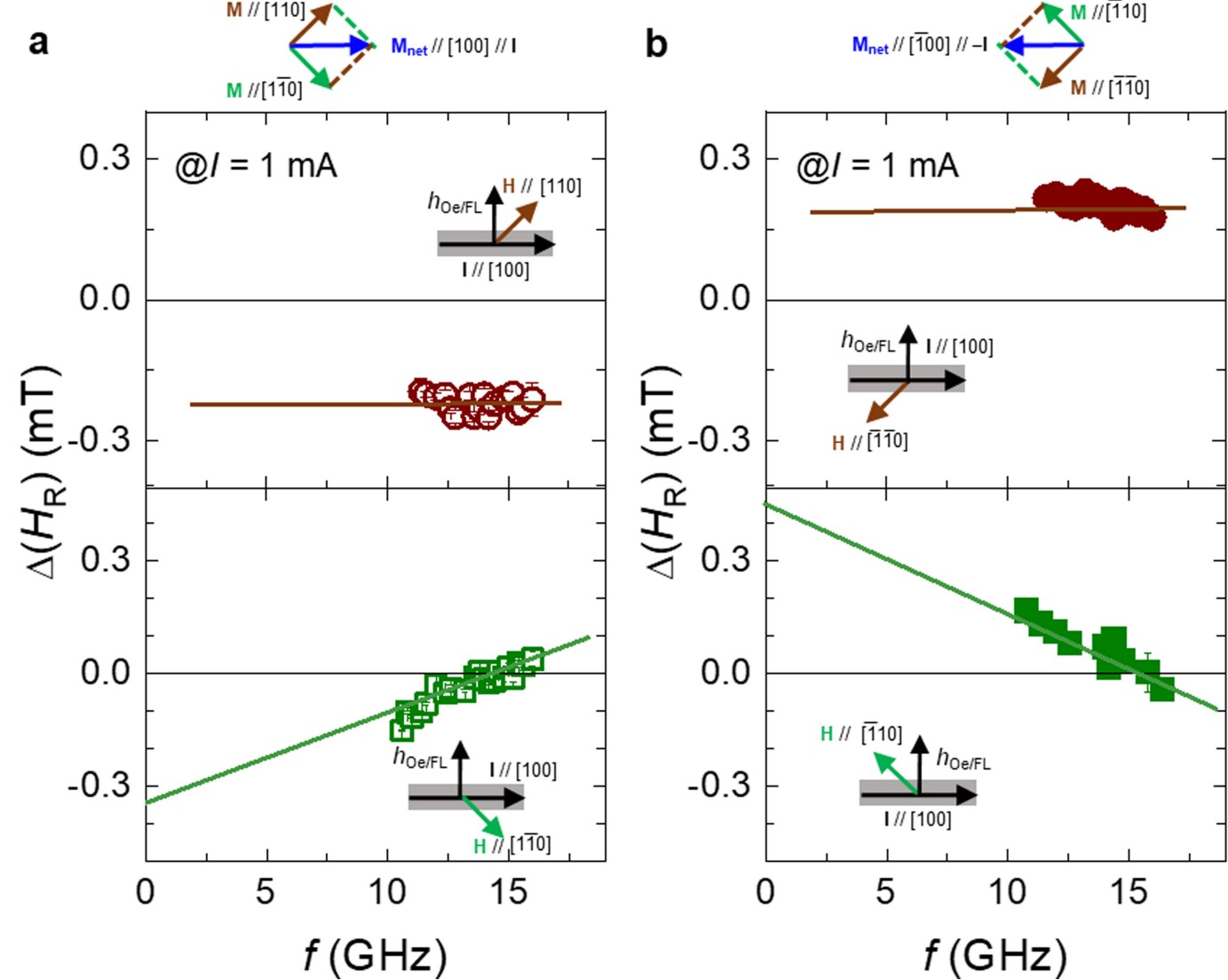

**Extended Data Fig. 9 | Shift of resonance field along easy and hard axes for** $t_{Fe}$ **= 1.2 nm. a**, Shift of the resonance field $\Delta H_R$ for $I = 1$ mA for **H** // **M** // [110] (easy axis) and **H** // **M** // [1$\bar{1}$0] (hard axis). **b**, Shift of the resonance field for $I = 1$ mA for **H** // **M** // [$\bar{1}\bar{1}$0] (easy axis) and **H** // **M** // [$\bar{1}$10] (hard axis) for the same sample. The inset in each figure shows the orientation of **H** with respect to the current. The upper panel of each figure shows the net magnetization, which is parallel to **I** for **a** and anti-parallel to **I** for **b**.

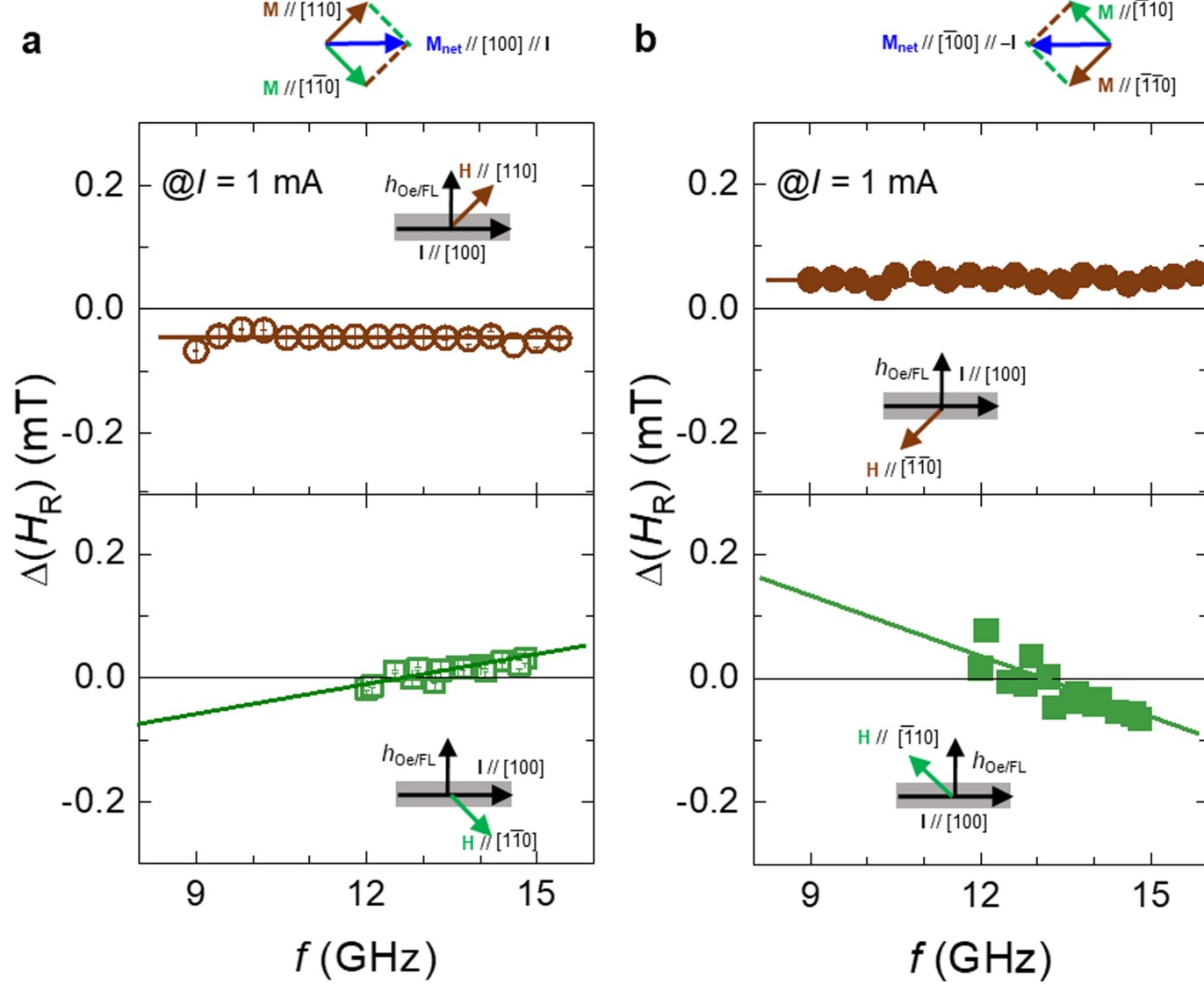

**Extended Data Fig. 10 | Shift of resonance field along easy and hard axes for** $t_{Fe}$ **= 2.2 nm. a**, Shift of the resonance field $\Delta H_R$ for $I$ = 1 mA for **H** // **M** // [110] (easy axis) and **H** // **M** // [1$\bar{1}$0] (hard axis). **b**, Shift of the resonance field for $I$ = 1 mA for **H** // **M** // [$\bar{1}\bar{1}$0] (easy axis) and **H** // **M** // [$\bar{1}$10] (hard axis) for the same sample. The inset in each figure shows the orientation of **H** with respect to the current. The upper panel of each figure shows the net magnetization, which is parallel to **I** for **a** and anti-parallel to **I** for **b**.

**Extended Data Table 1 | Summary of the $\Delta H_R$-$f$ relationships induced by $\Delta H_K$, $\Delta H_U$, and $\Delta H_B$ along easy and hard axes**

| Axis | $\Delta H_K$ | $\Delta H_B$ | $\Delta H_U$ |
|------|--------------|--------------|--------------|
| EA | $\Delta H_R = k_K f$ | $\Delta H_R = -\Delta H_B + k_B f$ | $\Delta H_R = \Delta H_U - k_U f$ |
| HA | $\Delta H_R = k_K f$ | $\Delta H_R = -\Delta H_B + k_B f$ | $\Delta H_R = -\Delta H_U$ |

**Extended Data Table 2 | Summary of $\Delta H_B$, $\Delta H_K$, $\Delta H_U$ and $h_{Oe/FL}$ for $t_{Fe}$ = 4.5 nm, 2.8 nm, 2.2 nm and 1.2 nm**

| | $t_{Fe}$ = 4.5 nm | | $t_{Fe}$ = 2.8 nm | | $t_{Fe}$ = 2.2 nm | | $t_{Fe}$ = 1.2 nm | |
|---|---|---|---|---|---|---|---|---|
| | **+M** | **−M** | **+M** | **−M** | **+M** | **−M** | **+M** | **−M** |
| $\Delta H_B$ (mT) | 0 | 0 | 0 | 0 | 0.11 | −0.14 | 0.26 | −0.26 |
| $\Delta H_K$ (mT) | 0 | 0 | 0 | 0 | 1.5 | −1.8 | 2.0 | −2.0 |
| $\Delta H_U$ (mT) | 0 | 0 | 0 | 0 | 1.6 | −1.9 | 2.5 | −2.5 |
| $|h_{Oe/FL}|$ (mT) | 0.03 | | 0.08 | | 1.56 | | 2.24 | |