## [Peer Review File · Nature]

Manuscript Title: Signatures of magnetism control by flow of angular momentum

Reviewer Comments & Author Rebuttals

Reviewer Reports on the Initial Version:

Referees' comments:

Referee #1 (Remarks to the Author):

In this work, the authors report experimental observation of a modification of magnetization magnitude and magnetocrystalline anisotropy by spin current. They use a typical spin-orbit torque setup consisting of a ferromagnetic layer interfaced with heavy metal. They observe a current-induced torque, which is well known, however on top of the torque they observe an effect of the current on "static" magnetic properties, i.e. the magnetization and the magnetocrystalline anisotropy. This is an interesting result, but I do not think the interpretation is as clear as the authors claim. Below is a list of my comments:

1. The authors claim that "there has been no explicit observation so far of successful spin current driven manipulation of the magnitude of M ". I do not think that is true. There has been observation of an increase of the magnetization due to superdiffusive spin currents driven by ultrafast demagnetization: [Rudolf et al., Nature Communications 3, 1037 (2012)].

2. The authors explain the observed data by a spin current flowing from Pt due to the spin-Hall effect, which increases the magnetization and this changes the anisotropy. To me this is a plausible interpretation, but I don't see how other effects can be excluded from the presented data. It is well known that many mechanisms can contribute to the spin-orbit torque apart from the spin-Hall effect. This includes, for example, the inverse spin-galvanic effect (also known as Edelstein or Rashba-Edelstein effect), spin-Hall effect in the ferromagnet itself, spin currents induced by interfaces or orbital effects (which could include many different effects). These effects could also in principle cause the observed phenomena. Some are related to spin currents, but not all. The inverse spin-galvanic effects is a locally generated spin density, for example. Without further experiments, I am not convinced that separating into different effects is possible and in general it is very difficult. This is not necessarily such a huge problem for the paper itself as in my opinion the results can be interesting regardless of the exact mechanism. But it is not possible to claim that the effect is due to spin currents and due to SHE from Pt without showing that it is the case.

3. Perhaps even more importantly, it is not clear to me what exactly can be directly concluded from the experiments. The authors claim they observe a modification of the magnetization magnitude, but has that been observed directly or is that just an interpretation of the results? Perhaps I'm missing something, but it seems to me that the authors do not actually directly observe a change in magnetization but rather a change in anisotropy. However, even the change in anisotropy is not so clear. It is really possible to conclude from the FMR experiments that the observed effect must be

due to a change of anisotropy? The authors model the effect of a spin current by a magnetic field, is it not possible that what they observe is simply an effect of an effective magnetic field due to the current oriented along the magnetization direction? Such an effective field is typically ignored in discussions of spin-orbit torque since it does not lead to a torque directly, but it will be present. Personally I am not convinced that from the presented data it is really possible to make the conclusion that the effect is due to changing of magnetostatic properties rather than due to effective field associated with the spin-orbit torque. It may not matter that much in practice since either way the effect is induced by current and so not really static, but if the authors want to claim that they observe a change of magnetization and anisotropy they need to show that it is really the case.

4. I don't really understand how is the estimate of the areal spin density calculated and I cannot find the formula $n_{SHE} = \mu_S \lambda N$ in Ref. 9. Could the authors explain where this comes from? I find it quite important especially since they find that the induced spin is quite large. I don't understand why the areal spin density is not simply the spin accumulation per area.

5. It is not completely clear to me where is the magnetic field oriented in the calculations. The authors mention $H \parallel z$ in the main text and in the supplementary material, but this does not make much sense to me since the easy axis is in-plane, so I assume that for the anisotropy calculations the authors apply the field in-plane. However, even then it is not entirely clear to me what they do. Is it correct that for the calculation of the anisotropy, the magnetization is put along the easy axis [110] and the magnetic field is put in the same or opposite direction? Since the anisotropy is calculated with the torque method my understanding is that the anisotropy is evaluated directly from this calculation so the energy for magnetization along [1-10] is not actually directly calculated. However, if it was evaluated, would this approach correspond to having the magnetic field still along [110] direction and only the magnetization rotated or would this correspond to rotating both the magnetization and the magnetic field?

6. Could the authors give the magnitude of the magnetic field that is necessary to induce the same anisotropy change in the calculations as is observed in the experiment? I could not find this anywhere in the text.

Overall, this is in my opinion an interesting work, however it is not very clear whether the interpretation given by the authors is justified, thus I would not recommend the manuscript to be published in this form. In my opinion the results are likely interesting by themselves even if the mechanism of what is happening is not entirely understood, but this should be clearly described in the manuscript or more justification should be given.

I'm also not entirely sure the work is of a wide enough interest that it would warrant a publication in Nature, though it is not possible to fully judge the manuscript in this form. Even though, the idea of manipulating the magnetization magnitude with spin currents has been demonstrated before, it was done in a different context and I'm not aware of any demonstration of modifications of anisotropy, thus the work is quite novel. The work is definitely also quite interesting and could provide a new perspective important for understanding spin-orbit torque devices. However, although it could be argued that the work demonstrates, in contrast to previous works, a modification of static

properties, I am not very sure this distinction is actually meaningful. Ultimately, everything they observe is induced by current and consequently also proportional to current, thus it is not really static. It is known that current-induced torques can modify various properties of magnetic materials, for example, the damping is modified by anti-damping torque. Although, I am not aware of works showing that spin-orbit torque can directly change anisotropy, fundamentally I do not think it is surprising that even “static” properties of magnetic materials may be modified by presence of spin-orbit torque or a spin current in general and that effective anisotropy will be modified by presence of spin-orbit torque.

Referee #2 (Remarks to the Author):

In the manuscript by L. Chen et al., the authors report a route to control the magnitude of magnetization via spin current. Most studies use spin current to perturb or switch the magnetization, here the authors propose that spin current can be used to also alter the strength of the magnetization via (de)populating the majority or minority spins in the energy band. They use ferromagnetic resonance to study the proposed effect. The experimental evidence the authors present is the dependence of the relative magnitude of HR (+I) and HR (-I) on the excitation frequency for 1.2-nm-thick Fe film. From the FMR results, they obtain the frequency-dependent change of magnetic anisotropy, which they argue only can result from the change of magnetization. While I found the experimental results and their interpretations novel and interesting, the evidence in my point of view is relatively weak to support their claimed physical mechanism. Therefore, I want to hold my recommendation if the authors can address the following questions.

1. The Kittel formula the authors used is based on magnetic anisotropy. Since FMR could directly measure magnetization, can the authors comment on why not analyze the change of magnetization directly from the FMR results?
 2. Spin polarization is not proportional to magnetization. If I understand correctly, the energy band of a magnetic material is primarily determined by the lattice structure instead of the magnetization. The link between spin polarization and magnetization is missing in the current manuscript. Can the authors provide convincing evidence or argument on why the change of spin population at the Fermi surface changes magnetization?
 3. If the claimed physical phenomena are universal, spin current changes the spin population at the Fermi surface, and thus influences the magnetization, then why there are so many restrictions to observe the effect, such as the limit of thickness, the crystallinity of the film, the uniaxial or biaxial anisotropy and 45-degree magnetic field in respect to the charge current? Such restrictions make the claimed effect to be something negligible when studying the spin current injection, and lower the impact of the proposed physical phenomena.
 4. Is it possible to directly measure the change of magnetization with applied charge current? If the change is about 0.2%, could it be reflected directly from longitudinal MOKE measurement?
- Overall, I consider the physics proposed in the manuscript novel and of general interest. However, the experimental evidence or arguments need to be strengthened to support their claim.

Referee #3 (Remarks to the Author):

The manuscript of “Spin current control of magnetism” by Chen and co-authors reports the modulation of in-plane uniaxial magnetic anisotropy in ultra-thin Fe layers grown on GaAs by injecting the electric current into the Pt/Al/Fe layers. In order to explain the anisotropy modulation, the authors consider the following plausible scenario: the Pt layer generates the transverse spin current due to the spin-Hall effect, and the spin current gives rise to not only the spin-transfer torque acting on the Fe magnetic moments, but also the enhancement or suppression of band splitting between majority and minority spins of Fe, which leads to the change in magnetic anisotropy. As the authors mentioned, “A well-established concept to date uses gate voltages to control magnetic properties”. From that point of view, the spin current control of magnetic properties reported in this work involves the remarkable novelty. In addition, their idea to use the Fe/GaAs system, which allows the authors to examine the effect of the spin current on in-plane uniaxial magnetic anisotropy, and the sophisticated measurement technique highlight the originality of their work. However, I would like to point out the critical technical issue involved in this work, which may overturn the possible scenario the authors suggested, and in the worst case change the conclusion.

The critical technical issue comes from the stacking of Pt / Al. The combination of Pt and Al is a well-known system showing the “self-propagating reactions near room temperature”. According to the paper of J. Appl. Phys. 124, 095105 (2018), the significant self-propagating reaction occurs even in the case of Pt / Al multilayers deposited at the substrate temperature not exceeding 55°C, and the Pt-Al alloy intermixing layer exists between the Pt and Al layers. The thickness of the Pt-Al intermixing layer is typically 10 nm.

Although I am not sure the detailed condition for depositing the Al and Pt layers in the present work, I suppose that the authors had a device fabrication step in which the thin film was heated up above 55°C, e.g. for resist-coating, which promotes the interdiffusion between the Al and Pt layers. Even for as-deposited films, the interdiffusion may occur.

What are the disadvantages due to the formation of Pt-Al layer? Since the thickness of Al layer is 1.5 nm, it is easy to imagine that there is no layered structure and only the Pt-Al intermixing layer is formed, i.e. the authors used the Pt-Al alloy/Fe/GaAs structure. In the authors’ original idea, they expected that the Al layer insertion prevents from the proximity magnetization effect due to the adjacent Fe layer. However, that role does not work because of no Al layer. Although the Pt-Al is one of the spin Hall materials showing the large spin Hall angle fortunately, the authors may not exclude the contribution of proximity magnetization in the Pt-Al layer. If the proximity magnetization appears, the situation becomes very complicated. How is the effect of current-induced spin accumulation on the proximity magnetization? The effect of spin current coming from the Fe layer on the proximity magnetization in Pt-Al? How the composition gradient in the Pt-Al layer does affect the spin current flow? etc... There are many questions, which need to be solved to claim the spin current control of magnetism. This work is no longer a straightforward experiment. I think that one possible way to solve the technical issue is that the authors should do the same experiment in the sample with Pt / Cu / Fe on GaAs.

Considering the concerns given above, I cannot recommend the publication.

Author Rebuttals to Initial Comments:

Response to Reviewer #1

"In this work, the authors report experimental observation of a modification of magnetization magnitude and magnetocrystalline anisotropy by spin current. They use a typical spin-orbit torque setup consisting of a ferromagnetic layer interfaced with heavy metal. They observe a current-induced torque, which is well known, however on top of the torque they observe an effect of the current on "static" magnetic properties, i.e. the magnetization and the magnetocrystalline anisotropy. This is an interesting result, but I do not think the interpretation is as clear as the authors claim."

Our response:

We would like to thank the Reviewer for critical reading of our manuscript as well as for the valuable comments to improve the manuscript, which we have taken into account in our revised manuscript.

Comment #1 of Reviewer #1:

"The authors claim that "there has been no explicit observation so far of successful spin current driven manipulation of the magnitude of M ". I do not think that is true. There has been observation of an increase of the magnetization due to superdiffusive spin currents driven by ultrafast demagnetization: [Rudolf et al., Nature Communications 3, 1037 (2012)]."

Our response:

We would like to thank the Reviewer for suggesting the work by Rudolf et al. In this work, the authors have shown that, in a magnetic Ni/Ru/Fe tri-layer where the two magnetization layers are exchange coupled, excitation using an ultrashort laser pulse generates a so-called super diffusive spin current into the Ni film, which transiently enhances (decreases) the magnetization of the Fe film when their magnetization directions are aligned parallel (antiparallel). We would like to point out that there are significant differences between Rudolf et al. and our work: i) Rudolf et al. show that the effect is limited to a low excitation level since the super diffusive spin current, relevant for the ultrafast demagnetization of Ni, saturates at high excitation power. In contrast, in our case, since the generation of the spin current is linear with the dc charge current, the modulation of magnetization and magnetic anisotropies

never saturates below the damage threshold current of the device. ii) The change of magnetization by super diffusive spin current is a transient process which lasts on timescales comparable to ultrafast demagnetization. In contrast, in our work, all the physical processes are “static” and the modulation of magnetism emerges (disappears) once the dc current is on (off), which behaves like a ‘static gate control’.

We would like to note that, during the preparation of the manuscript, we did not consider transient behavior as shown in Rudolf *et al.*. We have now cited this important work as Ref. 19 at the bottom of Page 3, and commented that:

"Previous work has shown that, in a magnetic Ni/Ru/Fe tri-layer where the two magnetization layers are exchange coupled, an ultrafast super diffusive spin current generated in Ni by using laser excitation transiently enhances (decreases) the magnetization of Fe when the two ferromagnetic layers are aligned parallel (antiparallel). However, this transient effect is limited to low optical excitations since the super diffusive spin current generated by ultrafast demagnetization saturates at high excitation power."

Comment #2 of Reviewer #1:

"The authors explain the observed data by a spin current flowing from Pt due to the spin-Hall effect, which increases the magnetization and this changes the anisotropy. To me this is a plausible interpretation, but I don't see how other effects can be excluded from the presented data. It is well known that many mechanisms can contribute to the spin-orbit torque apart from the spin-Hall effect. This includes, for example, the inverse spin-galvanic effect (also known as Edelstein or Rashba-Edelstein effect), spin-Hall effect in the ferromagnet itself, spin currents induced by interfaces or orbital effects (which could include many different effects). These effects could also in principle cause the observed phenomena. Some are related to spin currents, but not all. The inverse spin-galvanic effects is a locally generated spin density, for example. Without further experiments, I am not convinced that separating into different effects is possible and in general it is very difficult. This is not necessarily such a huge problem for the paper itself as in my opinion the results can be interesting regardless of the exact mechanism. But it is not possible to claim that the effect is due to spin currents and due to SHE from Pt without showing that it is the case."

Our response:

We would like to thank the Reviewer for the suggestions regarding the origin of the spin current.

Indeed, various effects may contribute to the charge to spin conversion, e.g., the spin Hall effect (SHE) and the inverse spin galvanic effect (ISGE) due to Bychkov-Rashba spin-orbit interaction. Both are related to the spin angular momentum (SAM), require spin-orbit interaction in the bulk (SHE) and at the interface (ISGE). Moreover, both SHE and ISGE are expected to generate field- and damping-like- spin-orbit torques acting on the ferromagnet in HM/FM bilayers.

Besides these SAM-effects, recent theoretical and experimental works have shown that the counterparts, i.e., the orbital Hall effect and orbital Rashba effect involving the orbital angular momentum (OAM), should also exist, which also generates spin-orbit torques acting on a ferromagnet. In contrast to SHE which needs sizeable spin-orbit interaction, the OHE is more common and does not require spin-orbit interaction and thus can occur in a wider range of materials. Therefore, one could exclude the SHE and validate the OHE by choosing light materials, such as Ti and Cr. For heavy materials such as Pt, the OHE and SHE could coexist or the spin Hall effect may even emerge as a by-product of the OHE resulting from the orbit-to-spin conversion due to spin-orbit interaction (G. Sala, et al. Phys. Rev. Mater. 4, 033037 (2022)). However, so far, it is difficult to separate these two effects.

As suggested by the Reviewer, to address the multiple origins of spin currents in general, we have changed the text on Page 4:

"The spin accumulation generated by a charge current I , e.g. by the spin Hall effect (SHE) in the heavy metal,"

to

"The spin accumulation generated by a charge current I , e.g. by strong spin-splitting of the energy band of ferromagnetic metals⁸, by the spin Hall effect (SHE)²³ and/or orbital Hall effect

(OHE)^{24,25} in the bulk as well as by spin Rashba-Edelstein effect (alternatively named as the inverse spin galvanic effect)^{26,27} and/or orbital Rashba-Edelstein effect²⁸⁻³⁰ at interfaces,".

Furthermore, we have added the following references:

24. Choi, Y. *et al.* Observation of the orbit Hall effect in a light metal Ti. *Nature* 619, 52 (2023).
25. Sala, G. and Gambardella, P. Giant orbital Hall effect and orbit-to-spin conversion in 3d, 5d, and 4f metallic heterostructures. *Phys. Rev. Mater.* 4, 033037 (2022).
26. Edelstein, V. M. Spin polarization of conduction electrons induced by electric current in two-dimensional asymmetric electron systems. *Solid. State. Commu.* 73, 233-235 (1990).
27. Gambardella, P. *et al.* Current-induced spin-orbit torques. *Phil. Trans. R. Soc. A* 369, 3175-3197 (2011).
28. Yoda, *et al.* Orbital Edelstein effect as a condensed-matter analog of Solenoids. *Nano. Lett.* 18, 916-920 (2018).
29. Salemi, *et al.* Orbitaly dominated Rashba-Edelstein effect in noncentrosymmetric antiferromagnets. *Nature Commun.* 10, 5381 (2019).
30. Ding, S. *et al.* Observation of the orbital Rashba-Edelstein magnetoresistance. *Phys. Rev. Lett.* 128, 067201 (2022).

Regarding the Pt/Al/Fe/GaAs multilayers used in this work, since the observed modification of the magnetic anisotropy is only connected to the damping-like spin-orbit torque, we focus on the discussion of the origin of the damping-like spin-orbit torque. Based on our previous work, we believe we can rule out a significant spin-Rashba contribution to the damping-like spin-orbit torque as well as for the modification of the magnetic anisotropy. In more detail:

Figure R1. (a) Temperature dependence of the damping-like effective magnetic-field h_{DL} at $j_{Pt} = 10^{11} \text{ Am}^{-2}$ for Pt(4 nm)/Co bi-layers with different Co thickness measured by the second harmonic longitudinal resistance method. The inset shows the linear dependence of h_{DL} on t_{Co}^{-1} for each temperature. (b) Temperature dependence of the field-like effective magnetic-field h_{FL} at $j_{Pt} = 10^{11} \text{ Am}^{-2}$. The figures are adapted from L. Chen et al. *Phys. Rev. B* 105, L020406 (2022).

- I) We have quantified the field-like torque and damping-like torque in Pt (4 nm)/Co bilayers as a function of Co thickness and temperature by second harmonic longitudinal resistance measurements. The T -dependence of h_{DL} and h_{FL} is summarized in Figures R1a and R1b. By varying t_{Co} , we find that h_{FL} changes sign upon increasing t_{Co} while $h_{DL} \sim t_{Co}^{-1}$. This suggests that h_{FL} originates from both ISGE and SHE (and/or OHE), but h_{DL} stems only from SHE (and/or OHE). Moreover, the generation of h_{DL} via ISGE can be further excluded due to the observation of an H -linear dependence of unidirectional magneto-resistance (L. Chen et al. *Phys. Rev. B* 105, L020406 (2022)).
- II) We have also quantified the magnitude of the damping-like torque induced by ISGE at the Fe/GaAs interface, which is about 0.3 mT for $j = 1 \times 10^{11} \text{ Am}^{-2}$ (L. Chen et al. *Nat. Commun.* 7, 13802 (2016)). This is about 10 times smaller than the damping-like spin-orbit torque generated in Pt/Co bilayers under the same magnitude of excitation (Fig. R1). Thus, damping-like torque induced by ISGE at the Fe/GaAs interface is negligible and can be ignored.

The above arguments exclude the damping-like torque originating from the spin-Rashba

effect. However, we cannot distinguish SHE and OHE by using Pt as the spin current source (G. Sala, et al. Phys. Rev. Mater. 4, 033037 (2022)). Therefore, we have changed the text on Page 5

"The Pt layer with strong spin-orbit interaction serves as the source of the spin current generated by the spin Hall effect²² and/or the orbital Hall effect²⁵,".

We have replaced "spin Hall effect (or SHE)" by "spin current" in the manuscript and in the Supplementary Material.

We have also modified the sentence on Page 7:

"The weaker damping-like torque, generated by the Bychkov-Rashba-like and Dresselhaus-like spin-orbit interactions at the Fe/GaAs interface, plays a negligible role in the linewidth modulation³⁵."

Comment #3 of Reviewer #1:

"Perhaps even more importantly, it is not clear to me what exactly can be directly concluded from the experiments. The authors claim they observe a modification of the magnetization magnitude, but has that been observed directly or is that just an interpretation of the results? Perhaps I'm missing something, but it seems to me that the authors do not actually directly observe a change in magnetization but rather a change in anisotropy. However, even the change in anisotropy is not so clear. It is really possible to conclude from the FMR experiments that the observed effect must be due to a change of anisotropy? The authors model the effect of a spin current by a magnetic field, is it not possible that what they observe is simply an effect of an effective magnetic field due to the current oriented along the magnetization direction? Such an effective field is typically ignored in discussions of spin-orbit torque since it does not lead to a torque directly, but it will be present. Personally I am not convinced that from the presented data it is really possible to make the conclusion that the effect is due to changing of magnetostatic properties rather than due to effective field associated with the spin-orbit torque. It may not matter that much in practice since either way the effect is induced by current and so not really static, but if the authors want to claim that they observe a change of magnetization and anisotropy they need to show that it is really the case."

Our response:

Although the modulation of the magnetization by the spin current is not directly measured by a magnetometer (such as a SQUID), the extracted magnetic anisotropy values from the FMR measurements also directly reflect the magnetic properties of the sample since FMR detects the magnetic energy landscape. We also show that the extracted effective demagnetization field H_K is mainly related to the magnetization M due to shape anisotropy, and thus the modulation of H_K confirms the modulation of the magnetization in the ultrathin regime. The data analysis (Supplementary Note 7) seems not transparent because of the existence of in-plane magnetic anisotropies for the Fe/GaAs system, but all the data can be well-analyzed by the modified Kittel formula with well-quantified values. The final results show that the modification effect decreases with increasing Fe thickness (Fig. 5), which proves again the validity of the method we used to extract the modulation amplitude of the magnetic anisotropies. Therefore, we firmly believe that both the experimental results and the data analysis are convincing.

It is worth to mention that the FMR method is utilized for several reasons: i) FMR has a higher sensitivity than the static magnetization measurements. ii) The FMR method, together with the angular- and frequency-dependent measurements, is a standard way to quantify the magnetization, magnetic anisotropies and Gilbert damping. iii) The damping-like- and field-like torques can be simultaneously determined in a single experiment, and thus one can establish the connection between damping-like torque and the modification of magnetic anisotropies in this study. iv) The Joule heating effect, which also alters the magnetic properties of Fe, can easily be excluded from the I -dependence of H_R .

The Reviewer also worries that the modulation of the magnetic anisotropy is due to current induced effective magnetic-fields, i.e., the Oersted field, a field-like spin-orbit field and a damping-like spin-orbit field. All these effects can be excluded because: i) The Oersted field and field-like spin-orbit field, which are induced by the dc current, only shift the magnitude of the resonance field and are frequency independent. Therefore, the Oersted field and the field-like spin-orbit field only contribute to the intercept in the f -dependence of the dH_R/dI

curves as shown in Fig. 4. ii) The damping-like spin-orbit field only contributes to the modulation of the FMR linewidth, i.e., the f -linear $d(\Delta H)/df$ curves as shown in Fig. S7. iii) The f -linear dH_R/df curves observed in thinner samples (Fig. 4) cannot be explained by only considering dc-current induced field-like spin-orbit fields, and can only be explained by the change of the magnetization and the magnetic anisotropies according to eq. 3.

To make our presentation clear, we have added the advantages of the FMR method in the methods section, explaining why such a small modification (0.2%) can be observed.

We have also added one sentence on Page 9 to clarify why the magnetic-field/magnetization is applied along the HA and EA to quantify the magnetic anisotropies:

"Aligning the magnetization along the axis of high symmetry (i.e., the hard axis and the easy axis) makes it easier to quantify the modification of the magnetic anisotropies since the magnetic dragging effect is absent⁴⁰."

Comment #4 of Reviewer #1:

"I don't really understand how is the estimate of the areal spin density calculated and I cannot find the formula $n_{SHE} = \mu_S \lambda N$ in Ref. 9. Could the authors explain where this comes from? I find it quite important especially since they find that the induced spin is quite large. I don't understand why the areal spin density is not simply the spin accumulation per area."

Our response:

The equation is obtained as follows:

On Page 16 of Ref. 9 (A. Manchon, et al. *Rev. Mod. Phys.* **91**, 035004 (2019).), and on the right column and below eq. 25, the spin chemical potential μ (in unit of Volt) is given by

$$\mu = S/eN \tag{R1}$$

where S is the density of the spin accumulation (in unit of $\frac{1}{m^3}$), e the elementary charge and N the density of states at Fermi level. The same equation can be also found in W. Chen, M. Sigrist, J. Sinova, and D. Manske, *Phys. Rev. Lett.* **115**, 217203 (2015).

In our presentation, the spin chemical potential is given in terms of energy (in unit of Coulomb·Volt), i.e.,

$$\mu = 2e\lambda\xi E \tanh\left[\frac{t_{pt}}{2\lambda}\right] \quad (\text{R2})$$

(see e.g., Y. T. Chen, *et al. Phys. Rev. B* **87**, 144411 (2013), and in this paper μ is also called spin accumulation, and that is why we named it in the same way in the previous manuscript).

Thus, areal spin accumulation n_S (in unit of $\frac{1}{\text{m}^2}$) should be obtained from eqs. R1 and R2 by multiplying the spin diffusion length, i.e.,

$$n_S = S\lambda = \mu\lambda N \quad (\text{R3})$$

Note that the elementary charge 'e' (in eq. R1) is already contained in μ (eq. R2), and does not show up in eq. R3.

To avoid misunderstanding and to make the presentation clearer, we have changed the description of eq. R2 on Page 15:

"The spin accumulation at the interface⁹..."

to

"The spin chemical potential at the interface⁹..."

We have also changed " n_{SHE} " to " n_S " in order not to address the origin of the spin current.

Comment #5 of Reviewer #1:

"It is not completely clear to me where is the magnetic field oriented in the calculations. The authors mention $H \parallel z$ in the main text and in the supplementary material, but this does not make much sense to me since the easy axis is in-plane, so I assume that for the anisotropy calculations the authors apply the field in-plane. However, even then it is not entirely clear to me what they do. Is it correct that for the calculation of the anisotropy, the magnetization is put along the easy axis [110] and the magnetic field is put in the same or opposite direction? Since the anisotropy is calculated with the torque method my understanding is that the anisotropy is evaluated directly from this calculation so the energy for magnetization along [1-10] is not actually directly calculated. However, if it was evaluated, would this approach correspond to having the magnetic field still along [110] direction and only the magnetization rotated or would this correspond to rotating both the magnetization and the

magnetic field?"

Our response:

We admit that this issue requires a more detailed description. A corresponding extension is added to the supplemental material with details on the magnetic torque calculations (at the beginning of Supplementary Note 9). To answer the questions of the Reviewer directly, we would like to describe briefly the main features of these calculations.

I) The in-plane uniaxial magnetic anisotropy of the system is primarily a consequence of the two-fold C_{2v} symmetry of the Fe/GaAs interface. The absolute value of the corresponding anisotropy energy depends on details of the electronic structure. In particular, it may be tuned by changing the exchange splitting of the Fe spin-up and spin-down states (or, equivalently, their local magnetic moment), e.g. via spin current injection as done in the present experiment. With this, the main aim of the accompanying calculations is to demonstrate the relationship between the Fe magnetic moment and the in-plane uniaxial magnetic anisotropy energy.

II) There are essentially two ways to determine the magnetic anisotropy energy – either via the total energy or via magnetic torque calculations relying on the magnetic force theorem. The Reviewer's questions are obviously based on the former type of calculations which is accurate and gives the MCA energy as a difference $E_{[110]} - E_{[1-10]}$, with both total energies calculated self-consistently. The torque method is based on calculations of the magnetic torque using a frozen potential and relies on the magnetic force theorem. This typically gives an error of 5-10%. However, the advantage of the torque method is that it can be efficiently used to calculate the uniaxial magnetic anisotropy energy (see, e.g., X. Wang, R. Wu, D. S. Wang, and A. J. Freeman, Phys. Rev. B 54,61 (1996)). In particular, dealing with the Fe/GaAs system, the energy difference, $E_{[110]} - E_{[1-10]}$, may be represented by the magnetic torque calculated for the magnetic moment rotated within the plane with respect to the [110] direction. In order to investigate the impact of a magnetic field on the magnetic anisotropy, a corresponding self-consistent potential is calculated only for the magnetic moment and magnetic field orientation along the z direction. Later on this potential is used to calculate

the torque on the rotated magnetic moment, giving access to the energy difference $E_{[110]} - E_{[1-10]}$ modified in the presence of the magnetic field oriented parallel to the magnetic moment.

To address this issue raised by the Reviewer, we have added more details concerning the torque calculations in the first paragraph of Supplementary Note 9.

Comment #6 of Reviewer #1:

"Could the authors give the magnitude of the magnetic field that is necessary to induce the same anisotropy change in the calculations as is observed in the experiment? I could not find this anywhere in the text."

Our response:

The corresponding discussions are added to the supplementary information together with an update of Fig. S18b to clarify these discussions. As it is shown, a magnetic field of about 1.5 T is required to induce the same UMA energy change as seen in experiment using an electric current of 1 mA in the Pt layer. For details see the extension added to Supplementary Note 9.

To address this comment, we have also added one sentence on Page 17 of the main text:

"Moreover, to model a change of ΔH_U of 2.5 mT for a dc current of 1 mA as observed in experiment, an equivalent magnetic-field of ~1.5 T is needed."

"Overall, this is in my opinion an interesting work, however it is not very clear whether the interpretation given by the authors is justified, thus I would not recommend the manuscript to be published in this form. In my opinion the results are likely interesting by themselves even if the mechanism of what is happening is not entirely understood, but this should be clearly described in the manuscript or more justification should be given."

Our response:

We would like to thank Reviewer #1 for the positive evaluation of our work. Following the

Reviewer's suggestion, we have done the following modifications of the manuscript to further improve our presentation:

- i) The multiple origins of the spin current have been clarified, and we avoid to discuss the specific origin of the spin current generated in Pt.**
- ii) We have added the advantages of the FMR method used in this study in the Method section.**
- iii) We have also measured the Pt/Cu/Fe/GaAs multilayers as suggested by Reviewer III, which shows consistent results as the Pt/Al/Fe/GaAs multilayers.**
- iv) The theoretical calculation has been explained in more detail.**

I'm also not entirely sure the work is of a wide enough interest that it would warrant a publication in Nature, though it is not possible to fully judge the manuscript in this form. Even though, the idea of manipulating the magnetization magnitude with spin currents has been demonstrated before, it was done in a different context and I'm not aware of any demonstration of modifications of anisotropy, thus the work is quite novel. The work is definitely also quite interesting and could provide a new perspective important for understanding spin-orbit torque devices. However, although it could be argued that the works demonstrates, in contrast to previous works, a modification of static properties, I am not very sure this distinction is actually meaningful. Ultimately, everything they observe is induced by current and consequently also proportional to current, thus it is not really static. It is known that current-induced torques can modify various properties of magnetic materials, for example, the damping is modified by anti-damping torque. Although, I am not aware of works showing that spin-orbit torque can directly change anisotropy, fundamentally I do not think it is surprising that even "static" properties of magnetic materials may be modified by presence of spin-orbit torque or a spin current in general and that effective anisotropy will be modified by presence of spin-orbit torque."

Our response:

We would like to thank Reviewer #1 for the positive evaluation of our work. As the Reviewer mentioned: "there is no demonstration of modification of anisotropy, thus the work is quite novel". We believe that this work does not only provide a new approach to manipulate the magnetic order parameter (e.g., the magnetization) but also deepens the understanding of

spin-torque phenomena. In particular, our results contradict the conventional view that the magnitude of the magnetisation remains fixed during the spin transfer process, as originally postulated by Berger and Slonczewski (Berger et al. Phys. Rev. B 54, 9353 (1996) & Slonczewski et al. J. Mag. Mag. Mater. 159, L1-L7 (1996)). We also anticipate analogous phenomena (such as modification of other key magnetic parameters by spin currents, e.g., Curie temperature, coercive force, etc.) and significantly larger modification amplitudes in alternative, more efficient spin current sources exploiting a wide range of spin-torque materials. As this method offers previously unavailable functionalities, we strongly believe that this manuscript would attract attention.

Finally, we would like to thank Reviewer #1 once again for his/her time as well as for the positive suggestions/comments. We hope that the issues raised in the report are adequately explained.

Response to Reviewer #2

"In the manuscript by L. Chen et al., the authors report a route to control the magnitude of magnetization via spin current. Most studies use spin current to perturb or switch the magnetization, here the authors propose that spin current can be used to also alter the strength of the magnetization via (de)populating the majority or minority spins in the energy band. They use ferromagnetic resonance to study the proposed effect. The experimental evidence the authors present is the dependence of the relative magnitude of HR (+I) and HR (-I) on the excitation frequency for 1.2-nm-thick Fe film. From the FMR results, they obtain the frequency-dependent change of magnetic anisotropy, which they argue only can result from the change of magnetization. While I found the experimental results and their interpretations novel and interesting, the evidence in my point of view is relatively weak to support their claimed physical mechanism. Therefore, I want to hold my recommendation if the authors can address the following questions."

Our response:

We thank the Reviewer for critical reading of our manuscript as well as for the thoughtful and valuable comments to improve the manuscript. We have considered them carefully in our revised manuscript.

Comment #1 of Reviewer #2:

"The Kittel formula the authors used is based on magnetic anisotropy. Since FMR could directly measure magnetization, can the authors comment on why not analyze the change of magnetization directly from the FMR results?"

Figure R2. Inverse Fe thickness t_{Fe}^{-1} dependence of the effective demagnetization field H_K for Pt/Al/Fe/GaAs (solid circles), AlO_x/Fe/GaAs (open circles) and the saturation magnetization (solid squares) for AlO_x/Fe/GaAs.

Our response:

Strictly speaking, the effective demagnetization field H_K extracted from FMR measurements is related to the saturation magnetization M (due to shape anisotropy of a ferromagnetic thin film) as well as a possible perpendicular anisotropy H_{\perp} stemming from interfaces (e.g., the Fe/GaAs interface or the Al/Fe interface due to interfacial spin-orbit interaction), and $H_K = M - H_{\perp}$ holds. Thus, FMR measurements in principle do not directly measure the magnetization for thin-films. To confirm that the observed changes of H_K are indeed related to the modulation of the magnetization, we have quantified the magnitude of H_{\perp} in our samples by measuring the magnitude of the saturation magnetization by SQUID. The obtained M values of the $\text{AlO}_x/\text{Fe}/\text{GaAs}$ samples, together with the obtained H_K values determined by FMR, are shown in Fig. R2, which shows that the magnitude of M is close to H_K for thin samples. This indicates that the magnitude of H_{\perp} is negligibly small. Therefore, we can say that the magnitude of H_K mostly reflects the magnetization for the samples investigated, and the observed changes of H_K (also H_U and H_B) are indeed related to the modulation of magnetization.

To clarify this point, we have added the saturation magnetization values in Fig. S4c, and have added a sentence on Page 5 in the Supplementary Material:

"By comparing the values of H_K and M , we confirm that the main contribution to H_K stems from the magnetization due to the demagnetization field of the thin film."

Comment #2 of Reviewer #2:

"Spin polarization is not proportional to magnetization. If I understand correctly, the energy band of a magnetic material is primarily determined by the lattice structure instead of the magnetization. The link between spin polarization and magnetization is missing in the current manuscript. Can the authors provide convincing evidence or argument on why the change of spin population at the Fermi surface changes magnetization?"

Our response:

We have proposed as a possible mechanism that the spin polarization influences the magnitude of the exchange interaction and then changes the magnetization. This has been mainly discussed on Page 4 and on Page 15-16 in the manuscript:

A pure spin current can be represented in terms of two electrons with opposite k-vector carrying up and down spin moments. The inflow of the spin-up electrons leads to an increase of the occupation of the spin-up states of Fe (here corresponding to M pointing in the up direction), while the outflow of spin-down electrons leads to a decrease of the occupation of the spin-down d-states. This process is schematically shown in Fig. 1a, but for simplicity the de-occupation of the spin-down d-state by the out-flow of spin-down electrons is not shown in this figure. The increase (decrease) of the occupation of the spin-up (spin-down) d-states leads to an enhanced exchange splitting of the majority- and minority-spin band, and then leads to an enhancement of the magnetization as well as the magnetic anisotropies.

Similarly, one can imagine the modification if the magnetization is flipped to the down direction (Fig. 1b) and if the spin polarization of spin current is reversed (Figs. 1c and 1d)

To make our presentation clearer, we have changed "splitting of the majority- and minority-spin band" to "exchange splitting of the majority- and minority-spin band".

We have added a sentence on Page 16 and 17:

"The increase (decrease) in the occupancy of the spin-up (spin-down) d-states leads to an enhanced exchange splitting of the majority- and minority-spin band, and then leads to an enhancement of the magnetization as well as the magnetic anisotropies."

"Moreover, to model a change of ΔH_U of 2.5 mT for a dc current of 1 mA as observed in experiment, an equivalent magnetic-field of ~1.5 T is needed."

Comment #3 of Reviewer #2:

"If the claimed physical phenomena are universal, spin current changes the spin population at the fermi surface, and thus influences the magnetization, then why there are so many restrictions to observe the effect, such as the limit of thickness, the crystallinity of the film, the uniaxial or biaxial

anisotropy and 45-degree magnetic field in respect to the charge current? Such restrictions make the claimed effect to be something negligible when studying the spin current injection, and lower the impact of the proposed physical phenomena."

Our response:

We would like to thank the Reviewer for this comment. Since the modulation of magnetic anisotropies by spin currents is an interfacial effect, the fundamental limit for this new approach is that the ferromagnet must be in the ultrathin regime, which is similar to the case of using electric-field control of magnetism in a capacitor structure.

We choose the FMR method to detect such a small modification, and it is worth to mention that FMR has various advantages: i) FMR has a higher sensitivity than static magnetization measurements. ii) The FMR method, together with angle and frequency dependency measurements, is a standard way to quantify the magnetization, magnetic anisotropies and Gilbert damping. iii) The damping-like- and field-like torques can be simultaneously quantified in a single measurement, and thus it is possible to make a connection between damping-like torques and the modification of magnetic anisotropies. iv) The Joule heating effect, which also alters the magnetic properties of Fe, can be easily excluded from the I -dependence of H_R .

On the other hand, FMR also limits the choices of samples. To see clear FMR spectra, the Gilbert damping should be reasonably low. However, realizing low Gilbert damping in heavy/ultrathin ferromagnet bilayers (down to sub-nanometer) is challenging for conventional spin-orbit devices. This is because extrinsic effects, such as the spin pumping effect, large interfacial spin-orbit interaction, two-magnon scattering, magnetic proximity effect and inter-diffusion of atoms, etc., drastically enhance the damping value of the ferromagnet. We choose the ultra-thin single crystalline Fe films grown on GaAs (001) substrates because of very low Gilbert damping values α in the sub-nanometer thickness regime ($\alpha = 0.0076$ for $t_{Fe} = 0.91$ nm), 1-2 orders smaller than the damping value of Pt/Co bilayers (e.g., see L. Zhu et al. Phys. Rev. Lett. 123, 057203 (2019).).

Moreover, the biaxial- and uniaxial in-plane magnetic anisotropies of the Fe/GaAs system are not a restriction for this experiment. In such a system, aligning the magnetization along the axis of high symmetry (i.e., the hard axis and the easy axis) makes it easier to quantify the anisotropies since the magnetic dragging effect is absent. In principle, one could also perform similar experiments using ultrathin Py/Pt bi-layers and we have considered this before. The advantage for polycrystalline Py is that one can avoid the in-plane biaxial- and uniaxial anisotropies and simplify the data analysis. However, the damping value of sub-nanometer Py, which is expected to be larger than 0.1 for $t_{\text{Py}} = 1$ nm according to Fig. R3, is too large to see a FMR spectrum, and we had to give up this sample choice for FMR detection.

Figure R3. Inverse Py thickness t_{Py}^{-1} dependence of damping in Pt/Py bi-layers. In the sub-nanometer regime of Py, the damping value is expected to be larger than 0.1, which is more than ~ 10 times larger than that of Pt/Al/Fe/GaAs multilayers.

We have added one sentence on Page 9 to clarify why the magnetic-field/magnetization is applied along HA and EA to quantify the magnetic anisotropies:

"Aligning the magnetization along the axis of high symmetry (i.e., the hard axis and easy axis) makes it easier to quantify the modification of the magnetic anisotropies since the magnetic dragging effect is absent⁴⁰".

To address that the observed phenomenon is universal and not limited to the Fe/GaAs system, we have added a sentence in the last paragraph:

"Although the modulation of magnetism is demonstrated using a single crystalline ferromagnet with in-plane anisotropies, this concept also applies to polycrystalline

ferromagnets, such as Py".

We have also added the advantages of the FMR method in the methods section, explaining why such a small modification (0.2%) can be observed.

Comment #4 of Reviewer #2:

"Is it possible to directly measure the change of magnetization with applied charge current? If the change is about 0.2%, could it be reflected directly from longitudinal MOKE measurement?"

Our response:

We would like to thank the Review for this suggestion. Actually, some experiments are already in our mind to see the modulation of magnetism (e.g., magnetization, magnetic anisotropies, Curie temperature, etc) by magneto-transport and magneto-optical methods.

In principle, such experiments seem transparent, and one needs to simply perform measurements by applying a proper charge current. However, there may be two technical obstacles. The first one is how to subtract the Joule heating effect induced by the applied current in the data analysis. For the FMR method, Joule heating can be easily extracted from the I -dependence of H_R . But we haven't done this for other measurements yet. The second obstacle may be the sensitivity of the experimental method for the detection of the change of the magnetization and the magnetic anisotropies. One could probably overcome this problem by choosing a low saturation ferromagnet (e.g., Pt/GaMnAs bilayers), ultra-thin ferromagnets contacted with a more efficient spin current source. We expect to see more alternative measurements in the future.

"Overall, I consider the physics proposed in the manuscript novel and of general interest. However, the experimental evidence or arguments need to be strengthened to support their claim."

We would like to thank Reviewer #2 once again for his/her time and for the positive comments of our work. We hope that the issues raised in the report are adequately addressed.

Response to Reviewer #3

"The manuscript of "Spin current control of magnetism" by Chen and co-authors reports the modulation of in-plane uniaxial magnetic anisotropy in ultra-thin Fe layers grown on GaAs by injecting the electric current into the Pt/Al/Fe layers. In order to explain the anisotropy modulation, the authors consider the following plausible scenario: the Pt layer generates the transverse spin current due to the spin-Hall effect, and the spin current gives rise to not only the spin-transfer torque acting on the Fe magnetic moments, but also the enhancement or suppression of band splitting between majority and minority spins of Fe, which leads to the change in magnetic anisotropy. As the authors mentioned, "A well-established concept to date uses gate voltages to control magnetic properties". From that point of view, the spin current control of magnetic properties reported in this work involves the remarkable novelty. In addition, their idea to use the Fe/GaAs system, which allows the authors to examine the effect of the spin current on in-plane uniaxial magnetic anisotropy, and the sophisticated measurement technique highlight the originality of their work. However, I would like to point out the critical technical issue involved in this work, which may overturn the possible scenario the authors suggested, and in the worst case change the conclusion."

We thank the Reviewer for his/her time for critical reading of our manuscript as well as for the thoughtful comments and suggestions.

"The critical technical issue comes from the stacking of Pt / Al. The combination of Pt and Al is a well-known system showing the "self-propagating reactions near room temperature". According to the paper of J. Appl. Phys. 124, 095105 (2018), the significant self-propagating reaction occurs even in the case of Pt / Al multilayers deposited at the substrate temperature not exceeding 55°C, and the Pt-Al alloy intermixing layer exists between the Pt and Al layers. The thickness of the Pt-Al intermixing layer is typically 10 nm. Although I am not sure the detailed condition for depositing the Al and Pt layers in the present work, I suppose that the authors had a device fabrication step in which the thin film was heated up above 55°C, e.g. for resist-coating, which promotes the interdiffusion between the Al and Pt layers. Even for as-deposited films, the interdiffusion may occur. What are the disadvantages due to the formation of Pt-Al layer? Since the thickness of Al layer is 1.5 nm, it is easy to imagine that there is no layered structure and only the Pt-Al intermixing layer is formed, i.e. the authors used the Pt-Al alloy/Fe/GaAs structure. In the authors' original idea, they expected that the

Al layer insertion prevents from the proximity magnetization effect due to the adjacent Fe layer. However, that role does not work because of no Al layer. Although the Pt-Al is one of the spin Hall materials showing the large spin Hall angle fortunately, the authors may not exclude the contribution of proximity magnetization in the Pt-Al layer. If the proximity magnetization appears, the situation becomes very complicated. How is the effect of current-induced spin accumulation on the proximity magnetization? The effect of spin current coming from the Fe layer on the proximity magnetization in Pt-Al? How the composition gradient in the Pt-Al layer does affect the spin current flow? etc... There are many questions, which need to be solved to claim the spin current control of magnetism. This work is no longer a straightforward experiment. I think that one possible way to solve the technical issue is that the authors should do the same experiment in the sample with Pt / Cu / Fe on GaAs. Considering the concerns given above, I cannot recommend the publication."

Our response:

We would like to thank the Reviewer for suggesting the paper (J. Appl. Phys. 124, 095105 (2018)), which shows that in amorphous Al/Pt multilayers, prepared by dc magnetron sputtering, significant intermixing of atoms can occur. If this would also be the case for our samples, Pt atoms would diffuse to the Al/Fe interface and could become magnetically polarized because of the magnetic proximity effect. We believe that we can largely rule out the magnetic proximity effect and that the magnetic proximity effect is irrelevant for the modification of magnetism by a spin current. This is because:

Figure R4 RHEED images taken after the growth of GaAs (a), Fe (b), Al (c) and Pt (d).

Figure R5 (a) HRTEM image of the Pt/Al/Fe/GaAs multilayer. (b) HAADF-STEM image. (c) EDX of each element.

- l) Figs. R4a-d show, respectively, the reflection high-energy electron diffraction (RHEED) images of GaAs, Fe, Al and Pt. Sharp streaks have been observed after the growth of each layer, which indicate the epitaxial growth mode as well as good surface (interface) flatness.

To further characterize the structural properties of our samples, we have carried out high resolution transmission electron microscopy (HRTEM) measurements. Note that, before the HRTEM measurements, the sample has been prebaked at 110 °C, which is the highest temperature for device fabrication. The HRTEM image shows (Fig. R5a) that all the layers are crystalline, due to the pseudomorphic growth (as also indicated by RHEED patterns as shown in Fig. R4) by molecular beam epitaxy at room temperature. To resolve the distribution of the atoms, a Z-contrast in the high-angle annular dark field (HAADF) scanning transmission electron microscopy (STEM) image is shown in Fig. R5b, and blurry boundaries

(white lines) for each layer can still be seen. Pt, Al, Fe, Ga and As elemental chemical maps (Fig. R5c) have also been acquired using energy-dispersive X-ray (EDX) spectroscopy, which shows that there is no significant diffusion of Pt atoms into Fe. Note that a tiny amount of Ga is also present in the Pt layer, which is generated by the cut of the sample into a lamina by focused-ion-beam. All results show that the good crystallinity of the Al layer prevents the diffusion of Pt towards Fe, and therefore the magnetic proximity effect can be largely blocked.

Figure R6 (a) φ_H -dependence of ΔH for GaAs/Fe(1.2 nm)/Al(1.5 nm)/Pt(6 nm) measured at $f = 13$ GHz. **(b)** φ_H -dependence of ΔH for GaAs/Fe(1.4 nm)/Pt(6 nm) measured at $f = 18$ GHz. The solid lines in **(a)** and **(b)** are fits using the indicated damping values. The sample without Al separation layer shows a ~ 5 times larger damping than the sample with Al.

- II) It is known that, in Pt/ferromagnet bi-layers without a separation layer, the magnetic proximity seems inevitable because Pt is close to the Stoner condition. The static and dynamic exchange coupling between the ferromagnet and the proximity magnetized Pt produces two effects which should be detectable in the FMR measurements (Y. Sun et al. *Phys. Rev. Lett.* **111**, 106601 (2013).): I) The static coupling, which is proportional to M , generates a torque acting on the ferromagnet and manifests itself as a shift of H_R . II) The dynamic coupling, which is proportional to dM/dt , plays a role for the enhancement of damping.

To show that the magnetic proximity effect can be largely avoided in the

Pt/Al/Fe/GaAs samples, we have prepared and measured Pt/Fe/GaAs samples without Al separation layer. As shown in Fig. R6, the GaAs/Fe/Pt sample without Al layer (Fig. R6b) has a ~ 5 times higher damping value than the GaAs/Fe/Al/Pt sample with Al layer (Fig. R6a). The much lower damping value in the GaAs/Fe/Al/Pt sample indicates that the Al insertion layer significantly reduces the magnetic proximity. Note that the damping value of the GaAs/Fe/Al/Pt sample is just slightly higher than that of a pure GaAs/Fe/ AlO_x sample (Fig. S3) due to the spin pumping effect. Therefore, we conclude that the Al separation layer indeed plays a significant role to minimize the magnetic proximity effect.

Fig. R7. I -dependence of H_R measured for a Pt(6 nm)/Cu(2 nm)/Fe(1.3 nm)/GaAs (001) sample at selected frequencies for H along the $[110]$ -axis (a), the $[\bar{1}10]$ -axis (b) the $[\bar{1}10]$ -axis (c) and the $[1\bar{1}0]$ -axis (d). The inset in each figures shows the respective orientation of the charge current and the magnetic-field (magnetization). The solid lines are fitted by eq. 2, and dH_R/dI is obtained. For H along the easy axis ($[110]$ and $[\bar{1}10]$), the relative amplitude of $H_R(-I)$ and $H_R(+I)$ is independent of f . While for H along the hard axis ($[\bar{1}10]$ and $[1\bar{1}0]$), the relative amplitude of $H_R(-I)$ and $H_R(+I)$ strongly depends on f .

Fig. R8. (a) Shift of the resonance field dH_R/dI for $H // M // [110]$ (easy axis) and $H // M // [1\bar{1}0]$ (hard axis) for Pt(6 nm)/Cu(2 nm)/Fe(1.3 nm)/GaAs (001) multilayer. (b) Shift of the resonance field for $H // M // [110]$ (easy axis) and $H // M // [1\bar{1}0]$ (hard axis). The inset in each figure shows the orientation of H with respect to the current.

	Pt/Cu/Fe/GaAs	
	+ \mathbf{M}	- \mathbf{M}
ΔH_B (mT)	0.28	-0.36
ΔH_K (mT)	1.12	-1.63
ΔH_U (mT)	1.48	-1.92
$ h_{Oe/FL} $ (mT)	1.52	

Table R1. Summary of ΔH_B , ΔH_K , ΔH_U and $h_{Oe/FL}$ for Pt(6 nm)/Cu(2 nm)/Fe (1.0 nm)/GaAs (001) multilayer under $I = 1$ mA.

III) To further prove that the magnetic proximity effect plays no role for the modification of magnetism by spin current, we have also studied a Pt(6 nm)/Cu (2 nm)/Fe (1.3 nm)/GaAs multilayer with Cu separation layer as suggested by the Reviewer. This is because the Cu interlayer significantly reduces the magnetic proximity effect between Pt and a ferromagnet (Y. Sun et al. *Phys. Rev. Lett.* **111**, 106601 (2013), H. Nakayama, et al. *Phys. Rev. Lett.* **110**, 206601 (2013), C. Du, et al. *Phys. Rev. Appl.* **1**, 044004 (2014), W. Amamou, et al. *Phys. Rev. Mater.* **2**, 011401 (2018).). It has also been shown (M. Caminale et al. *Phys. Rev. B* **94**, 014414 (2016).) that a 1-nm Cu separation layer is enough to eliminate the magnetic proximity

effect as evidenced by XMCD. Figures R7a and b respectively show the I -dependence of H_R measured at selected frequencies for H along the $[110]$ -axis and the $[\bar{1}\bar{1}0]$ -axis. One can see that $H_R(-I) > H_R(+I)$ for $H // [110]$ and $H_R(-I) < H_R(+I)$ for $H // [\bar{1}\bar{1}0]$. For both $[110]$ - and $[\bar{1}\bar{1}0]$ -orientations, the relative amplitude of $H_R(-I)$ and $H_R(+I)$ is independent of frequency. However, for H along the $[\bar{1}\bar{1}0]$ -axis and the $[1\bar{1}0]$ -axis as shown in Figures R7c and d, the relative amplitude of $H_R(-I)$ and $H_R(+I)$ strongly depends on frequency, similar to the results presented in Fig. 3 of the main text.

The I -dependence of the H_R traces are fitted by eq. 2, and the f -dependence of the $d(H_R)/df$ values is shown for each field orientation in Fig. R8. The sample with Cu separation shows the same results as the sample with Al separation, and the modification of the magnetic anisotropies (Table R1) quantitatively matches the results with Al separation (Table S3). Therefore, we confirm that the observation of the modification of magnetism by spin current is not related to the magnetic proximity effect.

- IV) Although the magnetic proximity effect has been largely avoided in our samples by inserting Al and Cu separation layers, we cannot completely exclude this effect. We would like to point out, however, that the magnetic proximity effect (if it exists) plays no role in the modification of magnetism by spin current. This is because the modification of magnetism is only related to the damping-like torque. It has been shown (e.g., L. Zhu et al. *Phys. Rev. B* 98, 134406 (2018).) that the magnetic proximity effect in heavy metal/ferromagnetic metal bilayers has no discernible influence on the magnitude of the current-induced spin-orbit torques, spin memory loss and spin backflow. Therefore, this furthermore confirms that the magnetic proximity effect does not affect the physics discussed in this work.

To address the magnetic proximity effect, we have added a separate section in the

supplementary material (Supplementary Note 8) to discuss the influence of magnetic proximity effect on the modification of magnetism by spin current, and the results of Pt(6 nm)/Fe(1.6 nm)/GaAs sample as well as the results of the Pt(6 nm)/Cu(2 nm)/Fe (1.3 nm)/GaAs (001) multilayer have been included. We have also added the RHEED and TEM images in Supplementary Note 1.

We have also added a sentence on Page 14 of the main text:

"and a possible magnetic proximity effect plays no role for the modification (Supplementary Note 8).".

We would like to thank Reviewer #3 once again for critically reading the manuscript and for providing valuable comments, which helped us a lot to improve the manuscript. We hope that the issues raised in the report are adequately addressed.

List of changes made in the revised manuscript and in the revised Supplemental Information:

All the changes have been highlighted in the revised manuscript and in the revised Supplemental Information.

Reviewer Reports on the First Revision:

Referees' comments:

Referee #1 (Remarks to the Author):

Although the authors give a lengthy and detailed response, unfortunately I don't really feel that the core issues I have raised have been addressed. Specifically:

1. Although the authors have added some discussion about other potential origins of the spin-orbit torque, the interpretation has not really changed. It's puzzling to me that on one hand the authors write that the "modification of dynamic and static properties" could be due to the orbital Hall effect, but at the same time they everywhere attribute the effect to a spin current (including in the title). The orbital Hall effect does not generate a spin current, but an orbital current. This could possibly be converted to a spin current, but not necessarily. It could also be converted to a spin accumulation rather than a spin current or in principle it could influence the magnetic order directly due to the equilibrium orbital moment.

The authors claim that they can exclude spin or orbital Rashba-Edelstein effects, but this is based on papers studying different systems and so is really not directly applicable. I also note that even if the origin of the spin-orbit torque is the spin-Hall effect, it's completely possible that the origin of the modification of the "static" properties is something else, I don't see any reason why these two effects would have to have the same origin.

I simply don't see any evidence in the manuscript that the effect they observe is due to a spin current. It is certainly a sensible explanation and to me it also seems like the most likely one, but I also believe that other effects could play a role. I don't think it's necessarily that important for the paper if the effect is due to a spin current, but if the authors want to claim that it is, they need to show that it is the case.

2. I'm also not really convinced by the argument that the change in resonance field the authors observe in the FMR experiments can directly be understood as a change of anisotropy. The authors argue that the field-like spin-orbit field only shifts the magnitude of the resonance field and the damping-like spin-orbit field changes only the linewidth. This may be true, but that's based on the assumption of the simplest form of the spin-orbit torque. In general, other terms can exist and the dependence of the spin-orbit torque on magnetization can be quite complicated, see e.g. [Nature Nanotechnology 8, pages 587–593 (2013)]. The authors argue that the FMR detects the magnetic energy landscape and I can believe that the experiments show a modification of this landscape by the spin-orbit torque, but whether this is really due to a change of magnetization and consequently of anisotropy is in my opinion not clear.

3. The authors still keep the claim in the manuscript that "there has been no explicit observation so far of successful spin current driven manipulation of the magnitude of M ", even though in the next sentence they added a

reference to a work that has done precisely that. The authors argue that this "transient effect is limited to low optical excitations since the super diffusive spin current saturates at high excitation power" and that may be true, but it definitely still constitutes an observation of spin current driven manipulation of the magnitude of M.

As it is, the manuscript contains claims that are not justified and thus I cannot recommend it to be published. I believe the results themselves can be of interest as they are, but the uncertainty of the interpretation and the assumptions that it is based on should be clearly reflected in the manuscript.

Referee #2 (Remarks to the Author):

In the revised manuscript and the response letter, the authors have fully addressed all my questions. Therefore, I could recommend the publication of this manuscript in Nature.

Referee #3 (Remarks to the Author):

I appreciate authors' great effort to carry out the TEM observation for the Pt-Al/Fe structure and the additional experiment using the new sample of Pt/Cu/Fe. The new experiment for the Pt/Cu/Fe structures successfully dispels my concern about the contribution of proximity magnetism in Pt. However, the TEM image for the Pt-Al/Fe structure definitely shows the significant interdiffusion between the Pt layer and the Al layer as I pointed out. No boundary (interface) is observed in HAADF-STEM [Fig. 5R(b)], and the Al signal is remarkably detected at the Pt layer position in EDX analysis [Fig. 5R(c)]. This is the strong evidence that the sample does not form the Pt/Al/Fe layered structure, but forms the Pt-Al alloy / Fe layer. Even in a little bit better case, that is Pt-Al alloy / Al / Fe. I don't want the authors to misunderstand my comment. I am never concerned about the diffusion of Pt into Fe. I did not point out such a concern in my former report. My concern is, although the authors claim the formation of Pt/Al/Fe layered structure and consider the spin Hall effect of Pt, in fact, the sample consists of Pt-Al alloy layer and Fe layer and spin Hall effect comes from Pt-Al.

The experimental fact (TEM images) says no layered structure of Pt and Al. The authors have already known this experimental fact. Nevertheless, they do not mention this point in this paper, and keep using the phrase of "Pt/Al/Fe". Although the TEM images were added in Supplementary Material as Supplementary Note 1, the main text does not cite this Supplementary Note 1. Why not?

Nature is the most influential journal, so if there is misinformation, many people will misunderstand and the research community will go in the wrong direction. If many people read this paper and misunderstand that Al is safe to use as a spacer layer adjacent to a Pt layer because it does not diffuse with Pt, there will be more erroneous reports of experiments that are not properly designed and will cause the community to regress.

I strongly recommend the authors to clearly mention the fact in the main text, that the Pt and Al are intermixed, and in the actual structure, a Pt-Al alloy layer is formed.

Author Rebuttals to First Revision:

Response to Reviewer #1

"Although the authors give a lengthy and detailed response, unfortunately I don't really feel that the core issues I have raised have been addressed. Specifically: "

We would like to thank the Reviewer for the careful review of the response and for the further comments. The Reviewer's insightful comments concerning the discussion of other possible mechanisms for the experimental observation, really helped us to further improve our manuscript. We will answer them point by point below.

Comment #1 of Reviewer #1:

"Although the authors have added some discussion about other potential origins of the spin-orbit torque, the interpretation has not really changed. It's puzzling to me that on one hand the authors write that the "modification of dynamic and static properties" could be due to the orbital Hall effect, but at the same time they everywhere attribute the effect to a spin current (including in the title). The orbital Hall effect does not generate a spin current, but an orbital current. This could possibly be converted to a spin current, but not necessarily. It could also be converted to a spin accumulation rather than a spin current or in principle it could influence the magnetic order directly due to the equilibrium orbital moment."

We agree with the reviewer's comments and in particular believe that we have misleadingly presented the general mechanisms for spin-orbit torques in the previous version. We have noticed that the oversimplified presentation does not help the reader. To make the presentation of spin-effect more accurate, we have changed the presentation in Page 4:

"The spin accumulation can be generated by strong spin-splitting of the energy band of ferromagnetic metals¹⁸, by the spin Hall effect⁶, by orbital Hall effect and subsequent conversion of the orbit current into a spin current via the spin-orbit interaction in the bulk⁷ as well as by spin Rashba-Edelstein effect (alternatively named inverse spin galvanic effect)^{8,9} at interfaces."

We have added a paragraph on Page 18 that addresses orbital effects as a possibility:

"In addition to the spin effect mentioned above, recent experimental and theoretical studies have shown that the orbital Hall effect¹⁰ and the orbital Rashba-Edelstein effect¹¹⁻¹³ can generate orbital angular momenta in the bulk of a nonmagnetic layer and at interfaces with broken inversion symmetry. The generated orbital momenta can exert a torque on the magnetization and could also cause a modification of the magnetization in two ways: I) the orbital current diffuses into an adjacent magnetic layer and is converted into a spin current by spin-orbit interaction^{14,15}. In this case the modification of the magnetization is analogous to the scenario discussed for a spin current. II) The orbital current could in principle act directly on the orbital part of the magnetization, generating orbit torques as well as leading to a modification of the orbital magnetization. The

change of the magnetization by orbital currents is expected to have the same odd symmetry as that induced by a spin current. Importantly, orbital effects could induce an even larger modification than spin effects because of the giant orbital Hall conductivity¹⁰ observed in some materials, and could affect thicker ferromagnets as it has been predicted that the orbital current dephasing length is longer than the spin dephasing length⁵⁹."

More importantly, we have changed the title to "Signatures of magnetism control by flow of angular momentum" in order not to address the spin current. We have also softened the abstract. In the new abstract, we mainly focus on the description of experimental facts, and the possibilities for the other mechanisms behind the experimental observation have also been discussed.

"The authors claim that they can exclude spin or orbital Rashba-Edelstein effects, but this is based on papers studying different systems and so is really not directly applicable. I also note that even if the origin of the spin-orbit torque is the spin-Hall effect, it's completely possible that the origin of the modification of the "static" properties is something else, I don't see any reason why these two effects would have to have the same origin."

"I simply don't see any evidence in the manuscript that the effect they observe is due to a spin current. It is certainly sensible explanation and to me it also seems like the most likely one, but I also believe that other effects could play a role. I don't think it's necessarily that important for the paper if the effect is due to a spin current, but if the authors want to claim that it is, they need to show that it is the case."

We would like to point out that we did not intend to "exclude spin or orbital Rashba-Edelstein effects". In our previous work (Phys. Rev. B 105, L020406 (2022)), we could show that the spin Hall effect in Pt contributes to the damping-like torque while both spin Hall effect and spin Rashba effect at the Co/Pt interface contribute to the field-like torque. Note that at that time we did not consider any orbital effects. This work simply confirms that Pt is a typical spin current source and that the spin current is responsible for the damping-like torque. A very recent work (Phys. Rev. Lett. 132, 236702 (2024)) further proves that Pt is a spin current source because: i) In rare-earth (RE) transition metal (TM)/Pt bilayers, the spin torque efficiency doesn't change significantly with the RE-TM ratio (Fig. 3). ii) The dependence of the spin torque efficiency on temperature is weak (Fig. 4), the reason being that the intrinsic spin Hall conductivity of Pt is not sensitive to temperature. Moreover, there is no experimental result demonstrating that Pt is not a spin current source as far as we know. Note that we try not to address the explicit mechanisms for spin current generation within Pt, i.e, i) a charge current generates an orbital current; the orbital current is converted into a spin current within Pt because of its large spin-orbit interaction. ii) The spin current is generated by the spin Hall effect. We avoid this because it is known that it is hard to distinguish these two processes.

Therefore, the spin current generated in the Pt layer plays a central role in the present measurements. We make this conclusion because the modification of the magnetic anisotropy is connected to the damping like

torque, which has been verified by the similar dependencies of modification of anisotropy (Fig. 5) and damping-like torque (Fig. S10) on Fe thickness. Moreover, since the amplitude of both the modification of the magnetic anisotropy and the damping-like torque depends inversely on Fe thickness, this indicates the experimental observation is a typical interfacial behavior. Therefore, we can also ignore the bulk effect from “*spin-Hall effect in the ferromagnet itself*” as suggested by the Reviewer in the first round.

Finally, we want to point out that the generation of a damping-like torque as well as the modification of magnetism shown in Fig. 1 requires a flow of spin, i.e., after the spin transfer process, the existing spin electrons, which is on average polarized along \mathbf{M} , enter the band and then change the magnetization. However, we are not sure if the spin accumulation generated by an interface effect could also change the magnetization or not. Although it is known that an interfacial spin accumulation, coupled to a ferromagnet via exchange interaction, can generate a damping-like torque (also called intrinsic SOT, see e.g., Nat. Nanotech. 9, 211-217 (2014), Nat. Commun. 7, 13802 (2016)). Therefore, we only addressed these effects in general.

To address that Pt is a spin current source, we have cited Ding, S. *et al. Phys. Rev. Lett.* 132, 236702 (2024).as Ref. 27.

Comment #2 of Reviewer #1:

"I'm also not really convinced by the argument that the change in resonance field the authors observe in the FMR experiments can directly be understood as a change of anisotropy. The authors argue that the field-like spin-orbit field only shifts the magnitude of the resonance field and the damping-like spin-orbit field changes only the linewidth. This may be true, but that's based on the assumption of the simplest form of the spin-orbit torque. In general, other terms can exist and the dependence of the spin-orbit torque on magnetization can be quite complicated, see e.g. [Nature Nanotechnology 8, pages 587–593 (2013)]. The authors argue that the FMR detects the magnetic energy landscape and I can believe that the experiments show a modification of this landscape by the spin-orbit torque, but whether this is really due a change of magnetization and consequently of anisotropy is in my opinion not clear."

Firstly, the angular dependence of the linewidth modulation (Fig. 2f) can be well fitted by conventional spin-orbit torques, i.e., eqs. S6 and S9, which is the T_0'' term ($\mathbf{m} \times \mathbf{y} \times \mathbf{m}$ term) of eq. 3 in K. Garello et al., Nature Nanotech. 8, 587 (2013). Obviously, the T_2'' [$(\mathbf{z} \times \mathbf{m})(\mathbf{m} \cdot \mathbf{x})$ term] and T_4'' [$(\mathbf{z} \times \mathbf{m})(\mathbf{m} \cdot \mathbf{x})(\mathbf{z} \times \mathbf{m})^2$ term] should give different angular dependences of linewidth modulation. Therefore, there is no need to consider these higher order SOTs.

Secondly, it is worth to address that the f -linear dH_R/dI curves observed in thinner samples (Fig. 4) cannot be explained by only considering dc-current induced field-like spin-orbit field and/or Oersted field, and can

only be explained by the change of the magnetic anisotropies according to eq. 3 (we cannot figure out a second explanation for this observation and the reviewer does not give a concrete hint what the explanation could be). The data analysis looks a little bit complicated but all the data [H_R , ΔH , dH_R/dI and $d(\Delta H)/dI$] can be well-quantified. Moreover, the modification effect decreases with increasing Fe thickness (Fig. 5), which shows that the modification of the static magnetic properties is an interfacial effect, which, in turn, is connected to the damping-like torque. This proves again the validity of the scenario which we propose in Fig. 1, and also shows that the modification of the FMR linewidth and the modification of the FMR resonance field are intimately related although the resonance field and linewidth are different physical quantities. Therefore, we firmly believe that our results are convincing and self-consistent.

We have added one sentence on Page 7:

"Since the angular dependence of the linewidth modulation can be well fitted by conventional spin-orbit torques^{19,39}, i.e., eqs. S6 and S9 in Supplementary Note 5, there is no need to consider other higher order SOTs⁴⁰",

and have added two more references:

39. Hayashi, M. *et al.* Quantitative characterization of the spin-orbit torque using harmonic Hall voltage measurements. *Phys. Rev. B* **89**, 144425 (2014).

40. Garello, K. *et al.* Symmetry and magnitude of spin-orbit torques in ferromagnetic heterostructures, *Nature Nanotech.* **8**, 587-593 (2013).

Comment #3 of Reviewer #1:

"The authors still keep the claim in the manuscript that "there has been no explicit observation so far of successful spin current driven manipulation of the magnitude of M", even though in the next sentence they added a reference to a work that has done precisely that. The authors argue that this "transient effect is limited to low optical excitations since the super diffusive spin current saturates at high excitation power" and that may be true, but it definitely still constitutes an observation of spin current driven manipulation of the magnitude of M."

We have changed our presentation from "there has been no explicit observation so far of ..." to "so far, there have been only few explicit observations of ..."

Comment #4 of Reviewer #1:

"As it is, the manuscript contains claims that are not justified and thus I cannot recommend it to be published. I believe the results themselves can be of interest as they are, but the uncertainty of the interpretation and the assumptions that it is based on should be clearly reflected in the manuscript."

We thank the Reviewer for the statement that "*I believe the results themselves can be of interest as they are*". As we explained above, we cannot figure out a second explanation for the experimental observation except by the modification of magnetic anisotropy. We believe the physical picture as well as the presentation of the experimental results are clearly interpreted. Since the modification of magnetism by spin current is a new effect, we believe the physics might not be fully explored and there might be other possible mechanisms, we would leave them as future work for deeper understandings.

Response to Reviewer #2

"In the revised manuscript and the response letter, the authors have fully addressed all my questions. Therefore, I could recommend the publication of this manuscript in Nature."

We would like to thank Reviewer #2 once again for his/her time as well as for his/her recommendation for publication.

Response to Reviewer #3

"I appreciate authors' great effort to carry out the TEM observation for the Pt-Al/Fe structure and the additional experiment using the new sample of Pt/Cu/Fe. The new experiment for the Pt/Cu/Fe structures successfully dispels my concern about the contribution of proximity magnetism in Pt. However, the TEM image for the Pt-Al/Fe structure definitely shows the significant interdiffusion between the Pt layer and the Al layer as I pointed out. No boundary (interface) is observed in HAADF-STEM [Fig. 5R(b)], and the Al signal is remarkably detected at the Pt layer position in EDX analysis [Fig. 5R(c)]. This is the strong evidence that the sample does not form the Pt/Al/Fe layered structure, but forms the Pt-Al alloy / Fe layer. Even in a little bit better case, that is Pt-Al alloy / Al / Fe. I don't want the authors to misunderstand my comment. I am never concerned about the diffusion of Pt into Fe. I did not point out such a concern in my former report. My concern is, although the authors claim the formation of Pt/Al/Fe layered structure and consider the spin Hall effect of Pt, in fact, the sample consists of Pt-Al alloy layer and Fe layer and spin Hall effect comes from Pt-Al.

The experimental fact (TEM images) says no layered structure of Pt and Al. The authors have already known this experimental fact. Nevertheless, they do not mention this point in this paper, and keep using the phrase of "Pt/Al/Fe". Although the TEM images were added in Supplementary Material as Supplementary Note 1, the main text does not cite this Supplementary Note 1. Why not? Nature is the most influential journal, so if there is misinformation, many people will misunderstand and the research community will go in the

wrong direction. If many people read this paper and misunderstand that Al is safe to use as a spacer layer adjacent to a Pt layer because it does not diffuse with Pt, there will be more erroneous reports of experiments that are not properly designed and will cause the community to regress.

I strongly recommend the authors to clearly mention the fact in the main text, that the Pt and Al are intermixed, and in the actual structure, a Pt-Al alloy layer is formed."

We would like to thank the Reviewer for this suggestion. We should have addressed the structure characterization in more detail in the manuscript. In our previous response, we partially misunderstood the Reviewer's comment and mainly focused on the magnetic proximity effect.

Following the Reviewer's suggestion, we have added the description of the structural properties on Page 5: "High resolution transmission electron microscopy measurements (Supplementary Note 1) show that there is diffusion of Al into Pt but no significant Al-Fe and Pt-Fe inter-diffusion. Therefore, the magnetic proximity effect between Fe and Pt is reduced. The intermixed Pt-Al alloy can be a good spin current generator^{6,15}; previous work²³ has shown that alloying Pt with Al enhances the spin-torque efficiency."

We have added two sentences in method section:

"Sharp reflection high-energy electron diffraction patterns have been observed after the growth of each layer (Supplementary Note 1), which indicate an epitaxial growth mode as well as good surface (interface) flatness. High resolution transmission electron microscopy measurements (Supplementary Note 1) show that i) all the layers are crystalline, ii) there is diffusion of Al into Pt but no significant Al-Fe and Pt-Fe inter-diffusion. Strictly speaking, the spin current is generated by the Al-Pt alloy but not by pure Pt."

We have also added two sentences in Supplementary Note 1:

"Although a significant diffusion of Al into Pt is observed in Fig. S2c, a sizeable spin-torque efficiency of the Pt-Al alloy (Supplementary Note 5) is still obtained. Previous work also showed that the spin-torque efficiency can be enhanced by alloying Pt with Al^{S1}."

List of changes made in the revised manuscript and in the revised Supplemental Information:

All the changes have been highlighted in the revised manuscript and in the revised Supplemental Information.

Reviewer Reports on the Second Revision:

Referees' comments:

Referee #1 (Remarks to the Author):

The authors have finally addressed the main comments. The manuscript is much improved in that the different possible mechanisms for the spin-orbit torque and the resonance field shift are discussed and importantly now a distinction is made between what is a direct experimental result (the resonance field shift) and what is an interpretation. This may seem obvious, but to a non-expert reader, it is very difficult to judge.

I'm still not convinced that the interpretation the authors give is necessarily the only one. The authors interpret the data based on a particular model, which can fit the data well, but it is possible that a more complicated model might fit the data as well. I also personally see no particular reason why the shift in the resonance field would necessarily have to have the same origin as the spin-orbit torque and I find it very likely that a spin accumulation with more complicated angular dependence than assumed here could also in principle explain the experimental findings. The argument that the shift in resonance field has a similar thickness dependence as the spin-orbit torque is not very convincing by itself since both are interfacial effects so this is expected. I also don't really understand why the authors claim that the change in anisotropy must be due to the change in magnetization. In my opinion, the anisotropy change (if the experiments indeed show that) could also have different origins. Thus overall I still don't feel that the interpretation the authors give is really very well justified.

I don't think this is some nitpicking either. In general, in my experience the origins of the spin-orbit torque can be quite complex. Competing mechanisms exist and effects that we now know can play a role, such as the interfacial spin currents or the orbital effects, have gone unnoticed for a long time. In my opinion, the possibility that the interpretation the authors give is in fact completely wrong is very real.

Nevertheless, ultimately it is the authors responsibility that their claims are substantiated and at least the manuscript now clearly states in the abstract what is an interpretation. It is also true that in this field it is quite common to give interpretations that are not very well substantiated and I have no reasons to doubt the experiments themselves. Thus I don't really want to object to the manuscript being published at this point.

As to whether the manuscript should be published in Nature, as I wrote in my original report, I'm not really convinced the work has a high enough impact for Nature. I don't see the distinction between static and dynamic properties so important since in reality nothing that is done here is really static, but is instead directly induced by the current, and vanishes without the current. Nevertheless, I don't really have a strong opinion on this and it is true that spin-orbit torques are overall a very important topic and this work potentially reveals some important aspects of the spin-orbit torque experiments that has been missed so far, thus an argument that the work belongs to Nature could also be made.

Referee #3 (Remarks to the Author):

I am satisfied with the revisions done by the authors. So, I am happy to recommend the publication in Nature.

Author Rebuttals to Second Revision:

Response to Reviewer #1

Comment of Reviewer #1:

"The authors have finally addressed the main comments. The manuscript is much improved in that the different possible mechanisms for the spin-orbit torque and the resonance field shift are discussed and importantly now a distinction is made between what is a direct experimental result (the resonance field shift) and what is an interpretation. This may seem obvious, but to a non-expert reader, it is very difficult to judge."

We are happy to hear from the Reviewer that: *"The manuscript is much improved in that..."* As we said before, the Reviewer's insightful comments indeed helped us to further improve our presentation as well as let us think deeper about our results. Therefore, we are very grateful for the Reviewer as well as for these important review processes.

"I'm still not convinced that the interpretation the authors give is necessarily the only one. The authors interpret the data based on a particular model, which can fit the data well, but it is possible that a more complicated model might fit the data as well. I also personally see no particular reason why the shift in the resonance field would necessarily have to have the same origin as the spin-orbit torque and I find it very likely that a spin accumulation with more complicated angular dependence than assumed here could also in principle explain the experimental findings. The argument that the shift in resonance field has a similar thickness dependence as the spin-orbit torque is not very convincing by itself since both are interfacial effects so this is expected. I also don't really understand why the authors claim that the change in anisotropy must be due to the change in magnetization. In my opinion, the anisotropy change (if the experiments indeed show that) could also have different origins. Thus overall I still don't feel that the interpretation the authors give is really very well justified."

Firstly, we would like to address that we cannot figure out a second explanation for the f -linear dH_R/dI curves observed in thinner samples (Fig. 4d) except by the modification of magnetic anisotropy. It is known that the starting point of FMR analysis is static magnetic energy landscape, which is related to the magnetic anisotropies. Therefore, it is natural to consider that the modification of magnetic anisotropy accounts for the f -linear dH_R/dI curves as observed in experiment. The data analysis looks a little bit complicated but all the data [H_R , ΔH , dH_R/dI and $d(\Delta H)/dI$] can be well-quantified self-consistently. The Reviewer commented that *"but it is possible that a more complicated model might fit the data as well."* Therefore, we have thought more carefully. One possibility that came to our mind could be a current induced modification of the Landé g -factor of Fe. In magnetic materials, it is known that g is related to the orbital moment μ_L and the spin moment μ_S , $g = \frac{2\mu_L}{\mu_S} + 2$.

A flow of spin- and orbit-angular momentum induced by a charge current could respectively modify the orbital- and spin-moment of Fe by $\Delta\mu_S$ and $\Delta\mu_L$, and then a change of the gyromagnetic ratio of Fe is expected. This could, in turn, lead to a shift of the FMR resonance fields, which linearly depends on frequency. However, if this were the case, an anisotropic modification of g is needed to interpret the data as observed experimentally (i.e., there is sizeable modification along the hard axis, but no modification along the easy axis). Since we cannot figure out why the modification of g could be anisotropic, we ignore this effect at the moment.

To address a current induced g -factor modification as a possible explanation to the data, we have added a paragraph in the Methods section to discuss alternative interpretation of the experimental results, and this is also referenced in the main text.

Secondly, the Reviewer also commented that *"a spin accumulation with more complicated angular dependence than assumed here could also in principle explain the experimental findings."* As we have pointed out in the second round that there is no need to consider higher order SOTs. This is because the angular dependence of the linewidth modulation (Fig. 2f) can be well explained by conventional spin-orbit torques. Actually, previous works also confirmed that there is no need to consider higher order terms for the linewidth modulation if Pt is used as a spin current source, e.g., Ref. 59: S. Kasai et al. Appl. Phys. Lett. 104, 092408 (2014). and Ref. 60: C. Safranski et al. Nature Nanotech. 14, 27-30 (2019).

Thirdly, we mentioned that *"the spin current induced modification of the magnetic energy landscape is of interfacial origin, similar to the damping-like spin-torque determined by ..."*. We want to say that this observation is unexpected and nontrivial to us. Based on this experimental fact, we see the connection between the modification and damping-like torque, and then we proposed that the possible mechanism for the modification, i.e, after the spin transfer process (generation of damping-like torques), the existing spin electrons, which is on average polarized along \mathbf{M} , enter the band and then change the magnetization.

Finally, we would like to point out that *the change in anisotropy is due to the change in magnetization*. This is because the magnitude of H_k (determined by FMR) is close to M (determined by magnetization measurements) for thin samples, which can be found in our response to Reviewer II in the first round. To address this point, we have also added a sentence in page 11, "Since $H_k \sim M$ in the ultra-thin regime (Methods), the change of magnetic anisotropy ΔH_k is directly related to the change of magnetization ΔM ".

"I don't think this is some nitpicking either. In general, in my experience the origins of the spin-orbit torque can be quite complex. Competing mechanisms exist and effects that we now know can play a role, such as the interfacial spin currents or the orbital effects, have gone unnoticed for a long time. In my opinion, the possibility that the interpretation the authors give is in fact completely wrong is very real."

"Nevertheless, ultimately it is the authors responsibility that their claims are substantiated and at least the manuscript now clearly states in the abstract what is an interpretation. It is also true that in this field it is quite common to give interpretations that are not very well substantiated and I have no reasons to doubt the experiments themselves. Thus I don't really want to object to the manuscript being published at this point."

We fully agree with the Reviewer that unknown mechanisms could play a role for the experimental observation, since "origins of the spin-orbit torque can be quite complex", and "such as the interfacial spin currents or the orbital effects, have gone unnoticed for a long time", as commented by the Reviewer. Therefore, we have changed our presentation by softening our argument, by discussing alternative interpretation of the data as well as by focusing mainly on the experimental results. We would leave the possible underlying mechanisms as future work for deeper understandings.

"As to whether the manuscript should be published in Nature, as I wrote in my original report, I'm not really convinced the work has a high enough impact for Nature. I don't see the distinction between static and dynamic properties so important since in reality nothing that is done here is really static, but is instead directly induced by the current, and vanishes without the current. Nevertheless, I don't really have a strong opinion on this and it is true that spin-orbit torques are overall a very important topic and this work potentially reveals some important aspects of the spin-orbit torque experiments that has been missed so far, thus an argument that the work belongs to Nature could also be made."

Finally, we would like to thank Reviewer #1 once again for his/her time as well as for insightful suggestions/comments. This indeed helps us a lot to improve our presentation. In the future, we expect to see similar effect as well as the modification of other key magnetic parameters by exploiting the wide range of spin-torque materials. As we mentioned before: this work does not only provide a new approach to manipulate the magnetic order parameter but also deepens the understanding of spin-torque phenomena. In particular, our results contradict the conventional view that the magnitude of the magnetisation remains fixed during the spin transfer process, as originally postulated by Berger and Slonczewski. As this method offers previously unavailable functionalities, we strongly believe that this manuscript would attract attention.

Response to Reviewer #3

"I am satisfied with the revisions done by the authors. So, I am happy to recommend the publication in Nature."

We would like to thank Reviewer #3 once again for his/her time as well as for his/her recommendation for publication.